# Pathology findings and correlation with body condition index in stranded killer whales (*Orcinus orca*) in the northeastern Pacific and Hawaii from 2004 to 2013

Stephen Raverty[1]*, Judy St. Leger[2], Dawn P. Noren[3], Kathy Burek Huntington[4], David S. Rotstein[5], Frances M. D. Gulland[6], John K. B. Ford[7], M. Bradley Hanson[3], Dyanna M. Lambourn[8], Jessie Huggins[9], Martha A. Delaney[10], Lisa Spaven[7], Teri Rowles[11], Lynne Barre[12], Paul Cottrell[13], Graeme Ellis[7], Tracey Goldstein[6], Karen Terio[10], Debbie Duffield[14], Jim Rice[15], Joseph K. Gaydos[16]

1 Animal Health Center, Ministry of Agriculture, Abbotsford, British Columbia, Canada, 2 Cornell University, Ithaca, New York, United States of America, 3 Conservation Biology Division, Northwest Fisheries Science Center, National Marine Fisheries Service, National Oceanic and Atmospheric Administration, Seattle, Washington, United States of America, 4 Alaska Veterinary Pathology Services, Eagle River, Alaska, United States of America, 5 Marine Mammal Pathology Service, Olney, Maryland, United States of America, 6 One Health Institute, School of Veterinary Medicine, University of California - Davis, Davis, California, United States of America, 7 Fisheries and Oceans Canada, Science Branch, Nanaimo, British Columbia, Canada, 8 Marine Mammal Investigations, Washington Department of Fish and Wildlife, Lakewood, Washington, United States of America, 9 Cascadia Research Collective, Olympia, Washington, United States of America, 10 Zoological Pathology Program, University of Illinois, Brookfield, Illinois, United States of America, 11 Office of Protected Resources, National Marine Fisheries Service, National Oceanic and Atmospheric Administration, Silver Spring, Maryland, United States of America, 12 West Coast Regional Office, National Marine Fisheries Service, National Oceanic and Atmospheric Administration, Seattle, Washington, United States of America, 13 Fisheries and Oceans Canada, Fisheries and Aquaculture Management, Vancouver, British Columbia, Canada, 14 Portland State University, Portland, Oregon, United States of America, 15 Oregon State University, Newport, Oregon, United States of America, 16 The SeaDoc Society, Karen C. Drayer Wildlife Health Center - Orcas Island Office, UC Davis School of Veterinary Medicine, Eastsound, Washington, United States of America

* Stephen.Raverty@gov.bc.ca

**Data Availability Statement:** All relevant data are within the manuscript.

## Abstract

Understanding health and mortality in killer whales (*Orcinus orca*) is crucial for management and conservation actions. We reviewed pathology reports from 53 animals that stranded in the eastern Pacific Ocean and Hawaii between 2004 and 2013 and used data from 35 animals that stranded from 2001 to 2017 to assess association with morphometrics, blubber thickness, body condition and cause of death. Of the 53 cases, cause of death was determined for 22 (42%) and nine additional animals demonstrated findings of significant importance for population health. Causes of calf mortalities included infectious disease, nutritional, and congenital malformations. Mortalities in sub-adults were due to trauma, malnutrition, and infectious disease and in adults due to bacterial infections, emaciation and blunt force trauma. Death related to human interaction was found in every age class. Important incidental findings included concurrent sarcocystosis and toxoplasmosis, uterine leiomyoma, vertebral periosteal proliferations, cookiecutter shark (*Isistius* sp.) bite wounds, excessive tooth wear and an ingested fish hook. Blubber thickness increased significantly

**Funding:** Funding for this investigation was provided by through NOAA (including contracts AB133F14SE2820, RA133F15SE1521, 1305M218CNFFK0068 and 1305M219PNFFK0366) and indirectly through numerous grants to marine mammal stranding programs from the John H. Prescott Marine Mammal Rescue Assistance Grant Program.

**Competing interests:** The authors have declared that no competing interests exist.

with body length (all $p < 0.001$). In contrast, there was no relationship between body length and an index of body condition (BCI). BCI was higher in animals that died from trauma. This study establishes a baseline for understanding health, nutritional status and causes of mortality in stranded killer whales. Given the evidence of direct human interactions on all age classes, in order to be most successful recovery efforts should address the threat of human interactions, especially for small endangered groups of killer whales that occur in close proximity to large human populations, interact with recreational and commercial fishers and transit established shipping lanes.

## Introduction

Killer whales (*Orcinus orca*) are a cosmopolitan species with an estimated total global abundance of over 55,000 animals [1]. Many populations exist, having geographically limited distributions that often are characterized by distinct dietary preference, vocalizations, genetics and even morphology [2]. As diet, behavior, regional anthropogenic activities, and environment conditions can impact health, understanding the health concerns for specific populations is important for management.

In the eastern North Pacific Ocean, three lineages or ecotypes of killer whales coexist: Resident (fish eaters), Transient (marine mammal eaters; also known as Bigg's), and Offshore (fish eating shark specialists) [2]. Within these lineages are four Resident populations, at least five distinct Transient populations, one Offshore population, and a single Hawaiian (USA) population that does not fall within one of the above ecotypes [2]. The northern resident population, listed by Canada as threatened, and the southern resident population listed by Canada and the USA as endangered, inhabit coastal and inland waters from southeastern Alaska (USA) to Monterey Bay, California (USA) with the outer coast of Washington State (USA) marking the southern extent of the northern resident range and Chatham Strait in southeast Alaska the northern limit for southern residents. Alaskan residents range from southeastern Alaska north to the Aleutian Islands and the Bering Sea. Transient whales are subdivided into at least five regional populations ranging from the Bering Sea to California. The offshore population inhabits waters along the continental shelf from the Aleutian Islands to California, and the Hawaiian population occurs in waters around the Hawaiian Islands [2].

Stranding patterns for killer whales in the North Pacific Ocean have been evaluated [3]. Examination of stranded animals provides a foundation for understanding killer whale natural history, diet, reproduction, anthropogenic stressors, presence of endemic and emerging diseases, and mortality patterns. Standardized examination of stranded animals using a killer whale necropsy protocol [4] is routinely performed, but necropsy findings have never been evaluated to understand causes and trends in mortality.

To date, data on killer whale health and disease are from captive animals [5–7] or case reports from free-ranging individuals or mass-stranded animals [8, 9]. An overview identified 15 infectious agents in killer whales and an additional 28 pathogens from sympatric cetacean species with a potential to affect killer whales [10]. Non-infectious health concerns include impacts from accumulated persistent organic pollutants (POPs) [11–13], human interactions including vessel collisions [14], interaction with fishing gear [15–17], the effects of noise [18–22], and consequences of reduced prey availability [22–24].

Understanding causes of wildlife morbidity and mortality allows identification of anthropogenic activities that can be mitigated [25]. Examples include the identification of vessel strike

as an important mortality factor in North Atlantic right whales (*Eubalaena glacialis)* [26–28] and human-caused increased kelp gull (*Larus dominicanus*) populations predating southern right whale calves (*Eubalaena australis)* [29, 30].

We retrospectively evaluated post-mortem examination records from killer whales that stranded over a ten-year period in the eastern Pacific Ocean and Hawaii to identify causes of morbidity and mortality.

## Materials and methods

Killer whale stranding response was permitted through multiple letters of authorization from NOAA to stranding networks in CA, OR, WA, AK, and HI. Records from killer whales that stranded in the North Pacific Ocean and Hawaii from January 2004 through December 2013 were reviewed. For each case, data included the stranding location and date, sex, ecotype or population, individual animal identity, age class, event history, post mortem images, nutritional condition, gross findings, morphologic diagnoses, and incidental findings. Ancillary diagnostic studies existed for some cases and included computed tomography (CT) scans, magnetic resonance imaging (MRI), viral, bacterial and fungal cultures, serologic assays and molecular screening for specific pathogens. Heavy metals and POPs were measured in some cases and when present, parasites were collected at necropsy and submitted for speciation.

Pathology reports from all necropsied killer whales were examined by four board-certified veterinary pathologists with extensive marine mammal experience and data were aggregated to identify significant findings by mutual consent (Table 1). Using this approach they identified proximate cause of death (COD) (the disease, injury, or process that initiated a sequence of events, which led to death) and ultimate COD (the final process that killed the animal), which were classified as congenital, environmental incident, euthanasia, infectious, inflammatory, metabolic, nutritional, traumatic (either resulting from a human interaction (HI) or non-HI), or unknown.

Environmental incidents reflect mortalities where whales were out of habitat (freshwater rivers) or mechanically stranded due to beach conformation and abrupt ebb tides. Metabolic cases represented instances where a catabolic derangement, such as hypoglycemia or electrolyte imbalance, did not appear to be associated with long-term nutritional limitation and was deemed the primary mortality factor. Nutritional cases included neonates with failure to thrive and older emaciated animals that did not exhibit evidence of underlying chronic disease. In some cases, a definitive diagnosis could be made (e.g. vessel strike for a traumatic COD). Occasionally important secondary pathologic findings were identified, but a proximate or ultimate COD could not be determined.

Photo-identification and morphometric data were used to classify animals as calves (includes fetus, neonate, and young calves), sub-adult, or adult. Fetuses included pre-term or aborted animals measuring less than 200 cm (straight length; measured from the tip of the rostrum to the deepest part of the fluke notch). Neonates were calves exhibiting structures such as fetal folds, vibrissae, colonic meconium or a patent umbilicus. Calves, including one fetus and three neonates, ranged from 201 to 360 cm long. Sub-adults were 361 to 500 cm, likely representing animals between 2 and 12 years of age and adults measured greater than 501 cm representing animals older than 12 years of age. Photo-identification, stomach content, or genetic sequencing of the mitochondrial control region (d-loop) [31] were used to identify the ecotype, population, and individual, when possible.

Body condition was evaluated using a subset of standard length, girth, and blubber thickness measurements from necropsy (see Fig 1) [4]. A body condition index (BCI) was calculated

**Table 1. Cause of death for killer whales examined from 2004–2013.**

| Orca ID | Necropsy Date | Location[1] | Lat / Long | Age Class[2] | Sex[3] | Population[4] | Proximate COD | Ultimate COD | Definitive Diagnosis | Secondary Pathology and Metazoan Parasites |
|---|---|---|---|---|---|---|---|---|---|---|
| 20040406 | 4/7/04 | HI | 20.826253/-156.809634 | A | F | U | Nutritional | Metabolic | Emaciation | Hepatitis, adrenalitis, endometritis, and myocardial fibrosis; stomach contained few nematodes (no ID) |
| 20040503 | 5/3/04 | OR | 43.102529/-124.433606 (est.) | A | F | T | Unknown | Unknown | Unknown | Leiomyoma (mesosalpinx) and cholangiohepatitis |
| 20050826 | 8/26/05 | AK | 58.47901/-135.99896 | C | F | AR (Likely AF or AG Pod) | Traumatic—HI | Infectious (bacterial) | Fishing interaction (halibut hook); septicemia and pharyngitis | *Anasakis simplex* in the stomach, proximal duodenum and lumen of the gall bladder; *Odhneriella subtila* in the small intestine |
| N018 | 12/2/05 | CA | 40.61533/-124.32919 | A | F | T | Traumatic—Unknown | Infectious (bacterial) | Trauma (Unknown origin) | Vertebral fracture with secondary infection; *Toxoplasma gondii* and *Sarcocystis neurona* |
| 20051207 | 12/7/05 | CA | 34.163889/-119.230278 | C | F | O | Infectious (bacterial) | Infectious (bacterial) | Septicemia; omphalophlebitis and omphaloarteritis | *Salmonella*—rib remodeling |
| L098 | 3/10/06 | BC | 49.6701 / -126.0718 (est.) | S | M | SR | Traumatic—HI | Metabolic | Trauma (vessel strike) | |
| C021 | 7/18/06 | BC | 54.3821 / -130.2680 (est.) | S | F | NR | Traumatic—HI | Metabolic | Trauma (vessel strike suspected) | |
| 20060728a | 8/2/06 | AK | 60.050833 / -144.180000 | A | U | T (GAT2 haplotype) | Environmental Incident | Metabolic | Beach casting | |
| 20060728b | 8/2/06 | AK | 60.050833 / -144.180000 | A | U | T (GAT2 haplotype) | Environmental Incident | Metabolic | Beach casting | |
| 20060804 | 8/4/06 | AK | 52.3712 / -175.9211 (est.) | C | U | U | Unknown | Unknown | Unknown | |
| 20061010 | 10/12/06 | AK | 60.7500 / -145.9833 (est.) | S | U | U | Unknown | Unknown | Unknown | |
| 20070520 | 5/20/07 | BC | 49.6590 / -126.8481 (est.) | A | F | T | Infectious (bacterial) | Infectious (bacterial) | Emaciation with peritonitis | Myocardial fibrosis; gastritis with occasional nematodes (no ID) |
| T086 | 5/22/07 | WA | 46.8534 / -124.1148 (est.) | A | F | T | Traumatic—HI | Metabolic | Trauma (vessel strike) | |
| 20070725 | 7/25/07 | AK | 59.53 / -149.38 | U | U | U | Unknown | Unknown | Unknown | |
| 20070802 | 8/2/07 | AK | 61.0323 / -146.7895 (est.) | C | U | U | Unknown | Unknown | Unknown | |
| 20080517 | 5/17/08 | AK | 57.61243 / -136.17065 | A | U | U | Unknown | Unknown | Unknown | |
| 20080524 | 5/24/08 | AK | 58.84510 / -160.21151 | U | U | U | Unknown | Unknown | Unknown | |
| 20080617 | 6/17/08 | AK | 59.0577 / -160.3758 (est.) | A | F | U | Unknown | Unknown | Unknown | |

(*Continued*)

**Table 1.** (Continued)

| Orca ID | Necropsy Date | Location[1] | Lat / Long | Age Class[2] | Sex[3] | Population[4] | Proximate COD | Ultimate COD | Definitive Diagnosis | Secondary Pathology and Metazoan Parasites |
|---|---|---|---|---|---|---|---|---|---|---|
| 20080726 | 7/26/08 | WA | 48.6167 / -123.1824 (est.) | F | U | SR | Unknown | Unknown | Abortion—cause Unknown | Aborted fetus -cause unknown |
| 20081021 | 10/22/08 | HI | 21.88883 / 27.01300 (est.) | S | F | U | Nutritional | Euthanasia | Emaciation | |
| T044 | 3/29/09 | BC | 50.9046 / -127.6718 (est.) | A | M | T | Unknown | Unknown | Unknown | |
| 20090405 | 4/5/09 | CA | 36.4776 / -121.9383 | C | F | T | Unknown | Unknown | Unknown | Necrotizing steatitis |
| 20090727 | 7/27/09 | AK | 52.87 / -172.89 (est.) | S | M | U | Unknown | Unknown | Unknown | |
| 20090802 | 8/2/09 | BC | 52.88 / -128.92 (est.) | A | F | T | Unknown | Unknown | Unknown | |
| 20090822 | 8/22/09 | AK | 59.832 / -147.606 (est.) | U | U | U | Unknown | Unknown | Unknown | |
| 20100306 | 3/6/10 | CA | 37.242311 / -122.418935 | N | F | U | Unknown | Unknown | Unknown | |
| 20100505 | 5/4/10 | BC | 48.397 / -123.982 (est.) | N | M | T | Nutritional | Metabolic | Failure to Thrive | |
| 20100614 | 6/14/10 | WA | 48.2025 / -124.6948 | A | F | T (haplotype 59) | Unknown | Unknown | Unknown | |
| 20100624 | 6/24/10 | AK | 63.694 / -170.478 (est.) | U | M | U | Unknown | Unknown | Unknown | |
| 20100715 | 7/15/10 | AK | 65.689697 / -168.536751 | A | U | U | Unknown | Unknown | Unknown | |
| 20100723 | 7/23/10 | AK | 65.093000 / -166.860333 | A | U | T (GAT haplotype) | Unknown | Unknown | Unknown | |
| 20100809 | 8/9/10 | AK | 63.694 / -170.478 (est.) | A | U | U | Unknown | Unknown | Unknown | |
| 20110223 | 2/23/11 | OR | 44.159299 / -124.118686 | S | U | T | Unknown | Unknown | Unknown | |
| 20110313 | 3/13/11 | AK | 57.080550 / -135.592033) | S | M | T | Inflammatory | Nutritional | Fibrinous peritonitis | Emaciation; Presumptive renal parasitic granuloma |
| 20110522 | 5/22/11 | AK | 59.301633 / -135.514576 (est) | C | M | NR | Congenital | Nutritional | Deformed jaw | Brachygnathia |
| 20111008a | 10/8/11 | AK | 58.921667 / -157.899444 | A | F | T | Environmental Incident | Nutritional | Extralimital event—inanition | |
| 20111008b | 10/8/11 | AK | 58.921667 / -157.899444 | A | F | T | Environmental Incident | Nutritional | Extralimital event—inanition | Severe tooth wear |
| 20111013 | 10/13/11 | AK | 58.921667 / -157.899444 | S | U | T | Environmental Incident | Nutritional | Extralimital event—inanition | |
| 20111114 | 11/14/11 | WA | 46.33240/-124.06710 | C | F | O | Congenital | Nutritional | Hiatal hernia | |
| O319 | 11/24/11 | CA | 38.194070 / -122.96524 | A | M | O | Traumatic—non-HI | Metabolic | Trauma (Unknown origin) | Severe tooth wear; 2–5 nematodes in stomach (no ID) |

(*Continued*)

**Table 1.** (Continued)

| Orca ID | Necropsy Date | Location[1] | Lat / Long | Age Class[2] | Sex[3] | Population[4] | Proximate COD | Ultimate COD | Definitive Diagnosis | Secondary Pathology and Metazoan Parasites |
|---------|---------------|-------------|------------|--------------|--------|---------------|---------------|--------------|----------------------|---------------------------------------------|
| L112 | 2/11/12 | WA | 46.4093 / -124.0613 | S | F | SR | Traumatic—Unknown | Metabolic | Trauma (blunt force, unknown origin) | Incomplete fusion of the dorsal process of C6 (spina bifida occulta); 30 non-embedded *Anisakis simplex* in fore stomach; *Crassicauda* spp. in the peribullar sinuses and fibrovenous plexus |
| 20120326 | 3/27/12 | AK | 56.671179 / -135.196338 | N | F | U | Nutritional | Metabolic | Failure to Thrive | |
| 20120508 | 5/8/12 | AK | 59.434 / -146.335 (est.) | C | U | U | Unknown | Unknown | Unknown | |
| 20120920 | 9/20/12 | AK | 59.546 / -139.727 (est.) | C | U | U | Unknown | Unknown | Unknown | |
| 20121001 | 10/1/12 | AK | 55.338 / -160.484 (est.) | A | F | U | Unknown | Unknown | Unknown | |
| 20130107 | 1/7/13 | WA | 48.1810 / -123.1156 | N | M | SR | Unknown | Metabolic | Failure to Thrive | Failure of passive immunity transfer |
| A011 | 1/10/13 | AK | 55/3633333 / -131.403333 | A | F | NR | Unknown | Unknown | Unknown | |
| 20130414 | 4/13/13 | BC | 48.608333 / -124.733333 | A | U | U | Unknown | Unknown | Unknown | |
| 20130904 | 9/4/13 | AK | 57.154381 / -170.198102 | N | F | U | Nutritional | Metabolic | Failure to Thrive | |
| I046 | 9/12/13 | BC | 49.0909 / -125.9364 (est.) | A | M | NR | Unknown | Unknown | Unknown | Minor tooth wear; colonic barbless fishing hook (treble); no evidence of perforation or mucosal ulceration |
| T171 | 10/18/13 | BC | 54.289 / -130.248 (est.) | A | F | T | Unknown | Unknown | Unknown | Severe boney remodeling of vertebral bodies |
| 20131104a | 11/7/13 | AK | 56.004 / -160.578 (est.) | A | F | O | Unknown | Unknown | Unknown | Severe tooth wear |
| 20131104b | 11/7/13 | AK | 56.004 / -160.578 (est.) | A | M | O (likely) | Unknown | Unknown | Unknown | Severe tooth wear |

[1] AK = Alaska, BC = British Columbia, CA = California, HI = Hawaii, OR = Oregon, WA = Washington

[2] A = Adult, C = Calf, F = Fetus, N = Neonate, S = Sub Adult, U = Unknown

[3] F = Female, M = Male, U = Unknown

[4] AR = Alaskan Resident, NR = Northern Resident, O = Offshore, SR = Southern Resident, T = Transient, U = Unknown

from two of these measurements:

$$BCI = \frac{body\ girth\ at\ the\ anterior\ dorsal\ fin\ insertion}{straight\ body\ length}$$

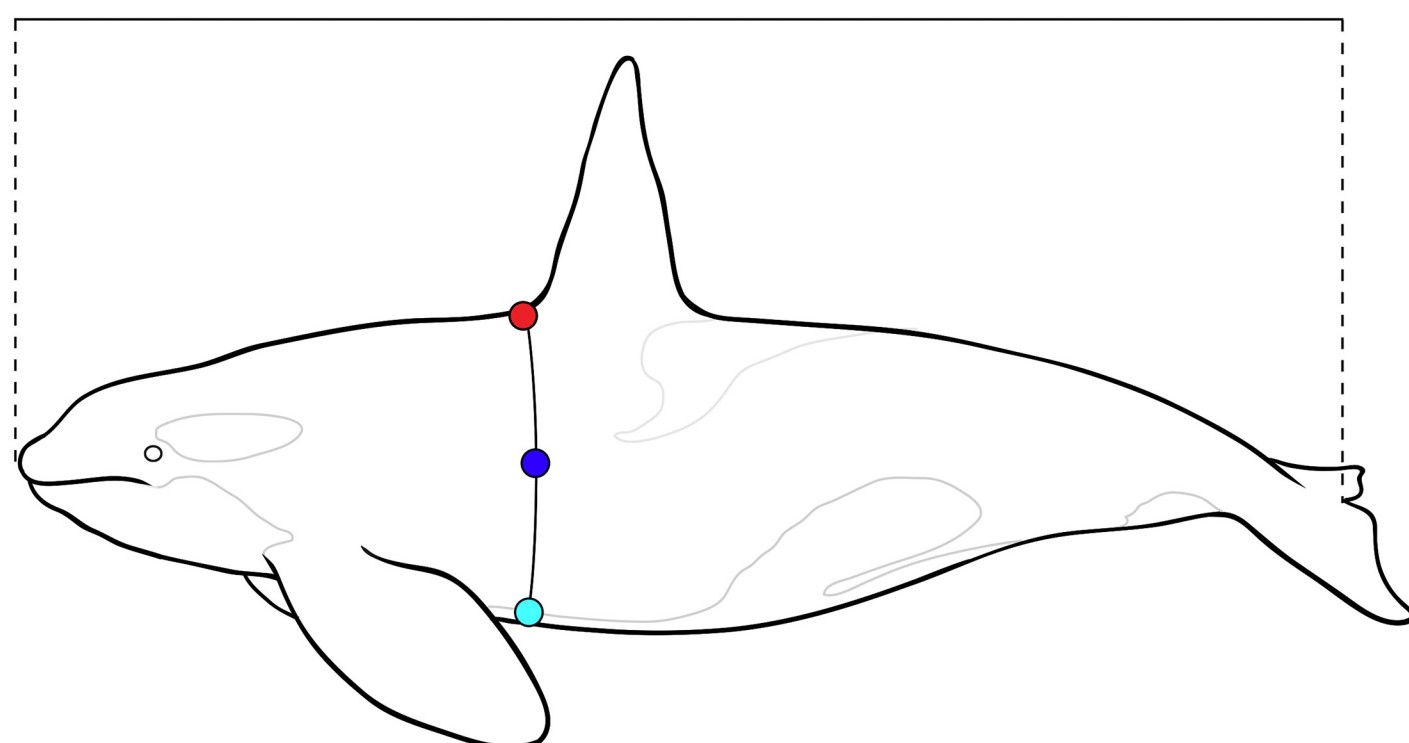

**Fig 1. Morphometric measurements taken from stranded killer whales at necropsy to assess body condition.** Total straight body length (solid line above whale), girth at the anterior insertion of the dorsal fin (solid line) and blubber thickness measured with a ruler at three sites: dorsal (red), lateral (dark blue), and ventral (light blue) are shown.

Location or position of the animal on the beach or partial immersion in water did not allow for morphometrics to be collected for all animals. We included data from 3 animals that stranded prior to 2004 and 15 that stranded after 2013 to have adequate sample sizes to assess the influence of body length and cause of death on blubber thickness and the body condition index (Table 2). A combined data set of measurements from 35 killer whales was used for body condition analyses. Due to incomplete necropsies, not all analyses include data from all 35 killer whales.

Killer whales are long-lived mammals with an extended growth period; thus, it is likely that blubber thickness and the BCI both change with age. In the absence of data on age for many of the stranded killer whales, body length was used as a proxy. Generalized linear models (GLM) were used to evaluate relationships between body length and blubber thickness measurements across the entire data set and by age class [fetus/neonates/calves (152.4 to 357.5 cm body length) and sub-adult/adult killer whales (375 to 792.5 cm body length)]. Based on the outcome of the GLM, ANCOVA or ANOVA was used, as appropriate, to determine differences in blubber thickness measurements across the three sites for the two age classes separately. Prior to performing these analyses, normality was confirmed by Shapiro-Wilk tests and equal variance was confirmed by Levene (for ANCOVA) or Brown-Forsythe (for ANOVA) tests. Because the relationship between body length and BCI is nonlinear, Spearman Rank Order Correlation was used to test for correlation between these two variables. Association between COD (e.g. trauma, infectious, or nutritional) and body condition (blubber thickness and BCI) was evaluated. Due to a small sample size, killer whales with infectious COD or nutritional COD were combined; both chronic infectious mortality and malnutrition were anticipated to

**Table 2. Additional stranded killer whales included for body condition index and cause of death analysis.**

| Orca ID | Stranding Date | Location[1] | Age Class[2] | Sex | Population | Ultimate Cause of Death |
|---|---|---|---|---|---|---|
| 20010506 | 5/6/01 | BC | N | U | T | Nutritional |
| CA189 | 1/2/02 | WA | A | F | T | Unknown |
| L060 | 4/16/02 | WA | A | F | SR | Trauma |
| 20140718 | 7/21/14 | AK | S | F | T | Unknown |
| J032 | 12/6/14 | BC | A | F | SR | Reproductive |
| J032 fetus | 12/6/14 | BC | F | U | SR | Reproductive |
| 20150418 | 4/8/15 | CA | A | M | T NT1 haplotype | Fishing Gear Entanglement |
| O059 | 10/15/15 | AK | A | F | O | Nutritional |
| 20151111 | 11/11/15 | AK | N | M | T | Nutritional |
| 20151224 | 12/24/15 | BC | N | F | T GAT1 haplotype | Nutritional |
| 20160323 | 3/23/16 | BC | N | F | SR | Trauma |
| L095 | 4/1/16 | BC | A | M | SR | Infectious |
| 20160410 | 4/10/16 | AK | N | M | U | Perinatal, Nutritional |
| 20160915 | 9/15/16 | BC | A | M | T | Trauma |
| J034 | 12/21/16 | BC | A | M | SR | Trauma |
| 20170220 | 2/20/17 | BC | C | M | T | Trauma |
| 20170612 | 6/12/17 | AK | A | M | AR Likely | Unknown, likely Nutritional |
| 20171227 | 12/27/17 | OR | C | F | T | Nutritional |

[1] AK = Alaska, BC = British Columbia, CA = California, OR = Oregon, WA = Washington

[2] A = Adult, C = Calf, F = Fetus, N = Neonate, S = Sub Adult

[3] F = Female, M = Male, U = Unknown

[4] AR = Alaskan Resident, O = Offshore, SR = Southern Resident, T = Transient, U = Unknown

be associated with poor body condition [32]. T-tests (normality confirmed by a Shapiro-Wilk test and equal variance confirmed by a Brown-Forsythe test) or Mann-Whitney Rank Sum Tests (when the normality test failed) were used to evaluate differences in body length, blubber thickness measurements, and BCI between killer whales that died from trauma and killer whales that died from infectious or nutritional CODs (combined). Because we had an *a priori* assumption that killer whales that died from trauma would have thicker blubber and higher BCI values than killer whales that died from infectious or nutritional causes [32], one-tailed hypothesis testing with T-tests were used for these comparisons when tests for normality and equal variance passed [33]. Otherwise, Mann-Whitney Rank Sum Tests were used,. A *P*-value of 0.05 was the critical statistical level of significance for all statistical tests. The generalized linear model analysis was conducted using 'stat' package in R Version 3.6.0. All other statistical and graphical analyses were conducted using SigmaPlot 14 Software (Systat Software, Inc., San Jose, CA, USA).

## Results

Over the 10-year period, 53 stranded killer whales were examined (Table 1). Of these, 22 (42%) had sufficient information to determine a cause of death. Significant ancillary diagnostic findings that were not considered to be associated with COD were identified in an additional seven animals. Likely due to some combination of length of the coast line [3], killer whale density, and detectability, 29 (55%) animals were from Alaska, nine (17%) from British Columbia (Canada), six (11%) from Washington, five (9%) from California, two (4%) in Oregon (USA) and two (4%) from Hawaii. Collectively, 62 significant pathologic findings were documented

from combined proximate COD, ultimate COD, and important secondary pathologic findings (Table 3). In 23 cases, the carcasses presented with either advanced autolysis or were poorly inaccessible and as a result limited data were collected providing insufficient opportunity to identify significant pathologic findings (Table 3).

## Case details by age class

**Neonates (neonatal calves).** A 245 kg neonate measuring 250cm (ID# 20120326) and a 203 cm long calf (ID# 20130904) were thin. Neonate 20130904 had serous atrophy of epicardial and mesenteric fat and no gastric contents. A third neonate, a 238 cm male transient (ID# 20100505), appeared to have been born alive, breathed, consumed a small quantity of colostrum and then died. Histopathology of multiple levels of the lung of this calf disclosed partially inflated alveolar spaces with a small number of widely scattered squames and a sparse meconium. In all three cases, the proximate COD was determined to be nutritional with an ultimate cause of death likely metabolic from hypoglycemia. None of these cases had associated data reflecting maternal condition.

**Calves (non-neonatal calves).** Four calf mortalities had significant findings. In two cases, the proximate cause of death was congenital. In the first case (ID# 20110522), a 289 cm northern resident calf died due to emaciation secondary to a cranio-facial malformation. There was brachygnathism with the mandible approximately 8 cm shorter than the maxilla (Fig 2). The second case (ID# 20111114) was an offshore calf, which had a hiatal hernia with partial entrapment of the second gastric chamber within the diaphragmatic defect.

One calf (ID# 20050826) died of sepsis, secondary to ingestion and impalement by a fish-hook. This Alaskan resident had a large halibut hook perforating the oropharynx (Fig 3). Additional findings included bronchopneumonia and necrotizing hepatitis. The traumatic perforation of the oropharynx was the proximate COD and emaciation in this case was considered secondary to the chronic inflammation, possibly compounded by difficulty swallowing. On gross examination, numerous superficial erosions, ulcerations and lacerations were also noted within the lingual, oropharyngeal and cranial esophageal mucosa, which were most likely related to the embedded fish hook, heavy gauge line and snap trailing from the lateral commissures of the mouth. The mixed bacterial population identified in the cellulitis were likely secondary opportunists, environmental contaminants or oral commensals; *Edwardsiella tarda* was isolated from a lymph node and likely accounted for the septicemia.

A young offshore calf that died of an infectious COD (ID# 20051207) presented with lymphadenopathy, omphaloarteritis and sepsis associated with *Salmonella* Newport [8]. Subsequent to publishing the case report [8], the skeleton from this case was cleaned and examined. Bilateral, firm, irregular nodular expansions of the costochondral junctions were noted in at least ten ribs. These areas were not examined histologically but were presumed to be costochondritis with bone remodeling related to bacteremia.

**Sub-adults.** Of the six sub-adult animals for which a COD was determined, three animals died with gross findings consistent with trauma. Two traumas were vessel strikes (northern resident ID# C021 and southern resident ID# L098) and one was blunt force trauma of unknown origin (southern resident ID# L112). Northern resident C021 was a thin, 470 cm sub-adult female with extensive cervical skeletal muscle hemorrhage, hemothorax, hemoabdomen and a subpleural hematoma of the left lung. Southern resident L098 was a 7-year-old socially isolated male that had no apparent interaction with other killer whales for 5 years. Witnesses reported that this habituated animal approached a vessel and was inadvertently drawn into the propeller. Despite intensive efforts, only a 31 kg portion of blubber, skin and pannicular skeletal muscle from the eye patch was recovered and the other remains of the carcass were

**Table 3. Sum of significant pathologic findings by age class for killer whales examined from 2004–2013.**

| Age Class | Abortion | Congenital | Degenerative | Environmental / Extralimital | Failure to thrive | Fisheries Interaction (hook) | Infection / Inflammation | Metabolic | Neoplasia | Nutritional | Trauma | Unknown |
|---|---|---|---|---|---|---|---|---|---|---|---|---|
| Adult | 0 | 0 | 5 | 4 | 0 | 1 | 4 | 5 | 1 | 4 | 3 | 11 |
| Sub–adult | 0 | 1 | 0 | 1 | 0 | 0 | 2 | 3 | 0 | 3 | 3 | 3 |
| Calf | 0 | 3 | 0 | 0 | 0 | 1 | 3 | 0 | 0 | 2 | 1 | 4 |
| Neonate | 0 | 0 | 0 | 0 | 4 | 0 | 0 | 4 | 0 | 3 | 0 | 1 |
| Fetus | 1 | 0 | 0 | 0 | 0 | 0 | 0 | 0 | 0 | 0 | 0 | 0 |
| Unknown | 0 | 0 | 0 | 0 | 0 | 0 | 0 | 0 | 0 | 0 | 0 | 4 |
| **Total** | **1** | **4** | **5** | **5** | **4** | **2** | **9** | **12** | **1** | **12** | **7** | **23** |

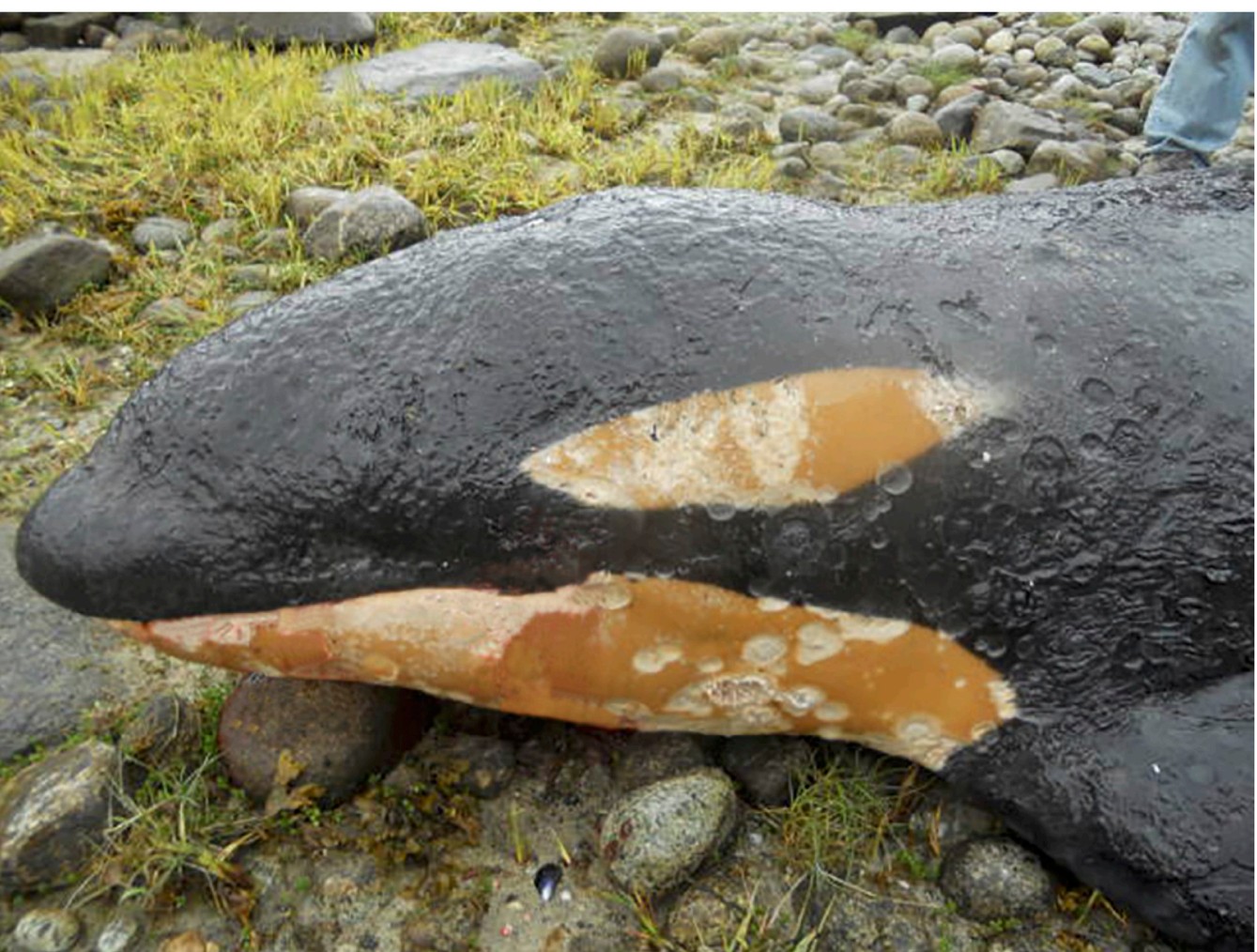

**Fig 2. A northern resident calf (ID# 20110522) that died of emaciation due to a congenital cranio-facial malformation.**

never found. Southern resident L112 had extensive subcutaneous bruising on the caudal head and neck, which extended deep into the adjoining epaxial musculature and tracked dependently along the hypodermis to the throat. On the right side of the carcass, the bruising extended to the anterior insertion of the pectoral fin.

For two sub-adult animals, the proximate COD was nutritional. The first, an emaciated female (ID# 20081021) that stranded alive in Hawaii, was subsequently euthanized. The nutritional condition was difficult to determine at the time of examination, but the pathologist's review of gross photos and collective necropsy findings concluded that emaciation/inanition was the most likely cause of stranding. Photographs revealed a significant depression behind the head at the location of the nuchal fat pad, prominent rib profiles in the thorax, and a "thin" tail stock (Fig 4). Blubber depth measured at the time of necropsy was 2.9–3.9cm. In addition to the poor nutritional condition, histopathology revealed moderate lymphoplasmacytic cholangiohepatitis with occasional bridging fibrosis, hemosiderosis and biliary ductular hyperplasia. As with the other animal that stranded in Hawaii (ID# 20040406), there was nonsuppurative adrenal adenitis and numerous irregular 3-5cm diameter cutaneous erosions and ulcerations, often infested with cyamid crustaceans (Fig 5). The thin Alaskan transient (ID#

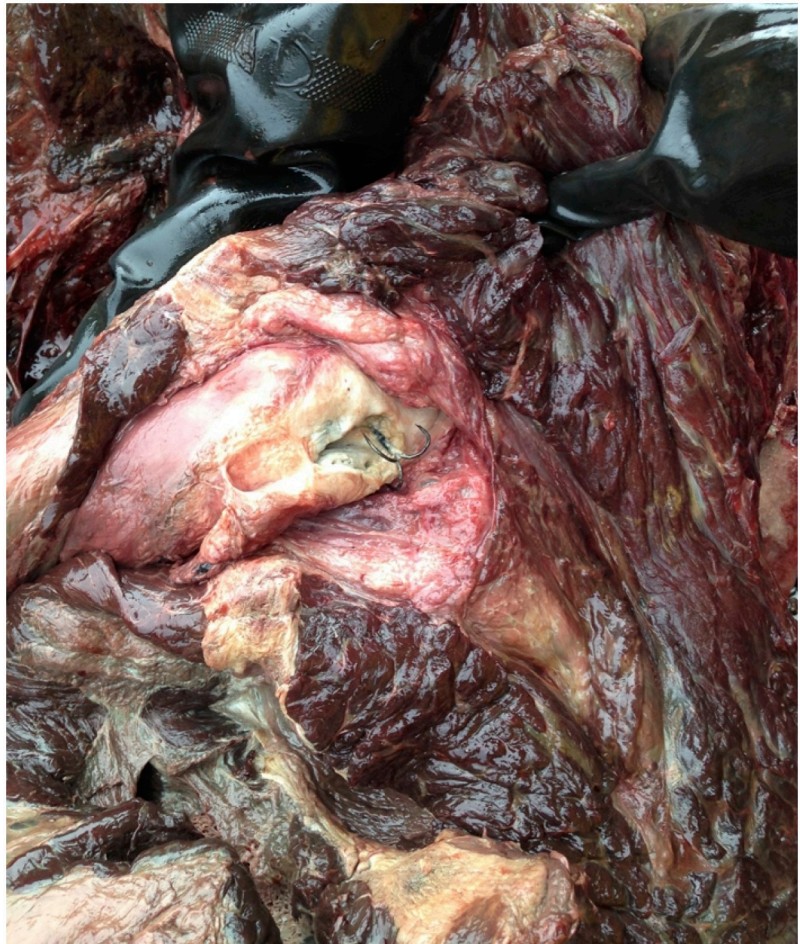

**Fig 3. Alaskan resident calf (ID# 20050826) with a halibut hook that perforated the wall of the oropharynx.** This animal died of sepsis resulting from the hook injury.

20111013) was a case of inanition subsequent to a period of six weeks or more up the Nushagak River. The specifics of this case are presented in the adult mortality section because that animal was one of three in a mass stranding event.

The sixth sub-adult animal was an emaciated transient male from Alaska (ID# 20110313) that died from peritonitis of undetermined etiology. Grossly, abdominal serosal adhesions were prominent and likely impeded normal intestinal motility, digestion and perhaps locomotion. The ultimate COD was nutritional.

**Adults.** A COD was assigned to nine adult stranded killer whales. Of these, three mortalities were classified as traumatic, including one case where mortality was associated with secondary abscessation. Additionally, two adults died from sequelae associated with protracted extralimital environmental incidents, two were attributed to natural environmental incidents (beach casting) and two cases were nutritional.

In 2007, a female transient (ID# T086) succumbed to catastrophic propeller strike. The carcass was not recovered but the intact dorsal fin, and a portion of the saddle patch were recovered on the beach (Fig 6). The deep parallel serrated incisions, abrupt, stepwise or repeating angulated margins and shearing of the blubber from the hypodermis characteristic of propeller strike formed the margins of the retrieved tissue [34]. The second adult that died from trauma,

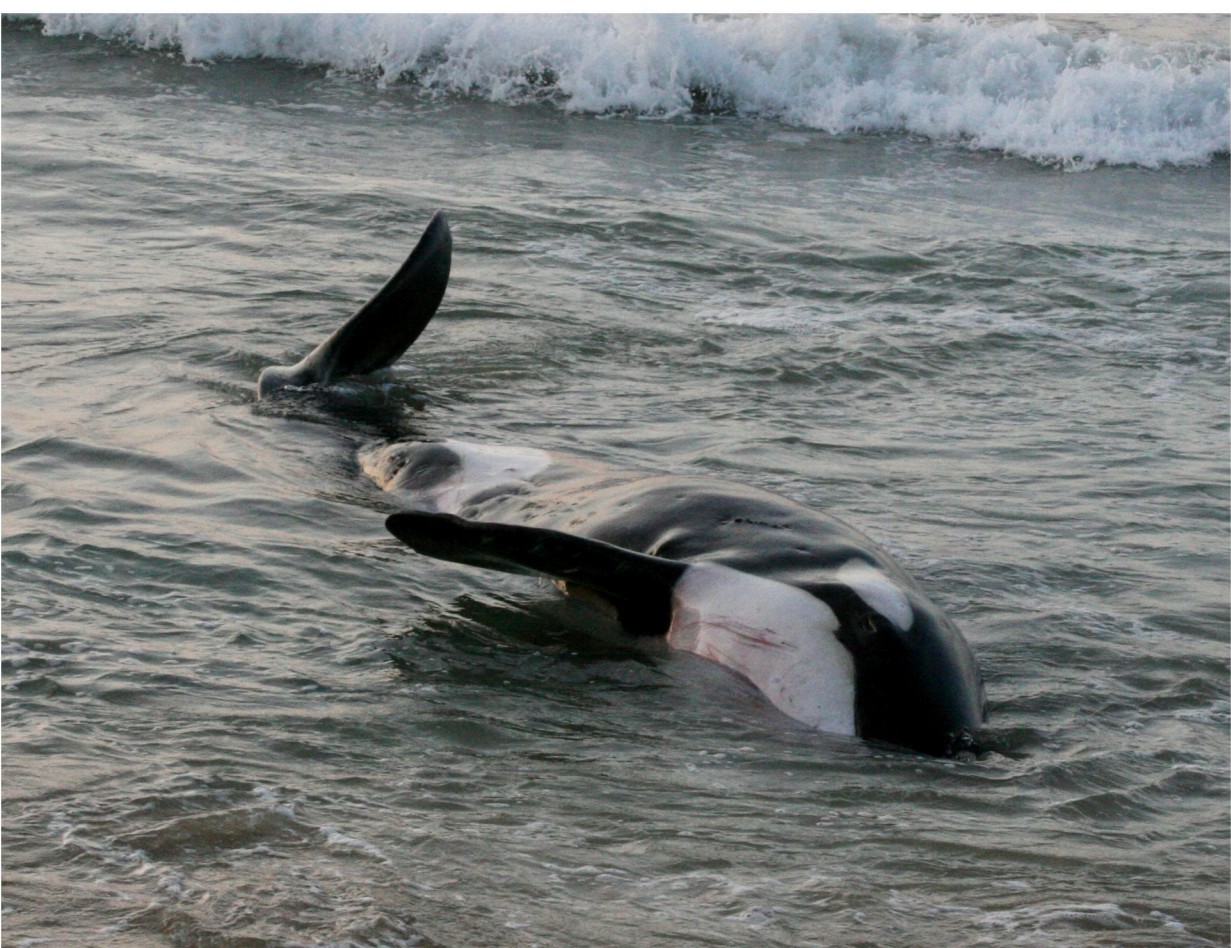

**Fig 4. Emaciated female killer whale (ID# 20081021) from Hawaii.** Note the depression behind the head at the location of the nuchal fat pad, prominent rib profiles in the thorax, and a thin tail stock.

an offshore male (ID# O319), had moderate hemorrhage within the musculature of the left thoracic wall and an acute transverse closed simple fracture of the left rib with associated hemothorax. There was also a pre-existing healed fracture of the same rib located approximately 12cm ventral to the fatal fracture. A transient female (ID# N018) necropsied in California had an extensive intramuscular bacterial abscess that included the vertebral bodies of the caudal peduncle. The infection was associated with a prior traumatic wound and secondary microbial contamination.

Environmental incidents were attributed as a proximate COD in 4 adult orcas. Two transient killer whales (ID# 20060728a and b) that stranded near Cordova, Alaska were likely feeding in shallow water when the ebb tide abruptly receded. A necropsy exam was complete in one animal and due to human safety concerns, was cursory in the second whale. In both cases, the animals were in good nutritional condition with no lesions suggesting co-morbidity or mortality beyond the environmental incident. The second environmental event occurred in 2011 when two transient adults (ID# 20111008a and b) and a sub-adult (ID# 20111013) swam up the Nushagak River from Bristol Bay near the town of Dillingham, Alaska. The animals were monitored in the freshwater river for six weeks before there was an abrupt drop in water level at which time the whales apparently live stranded on a sand bar and subsequently died.

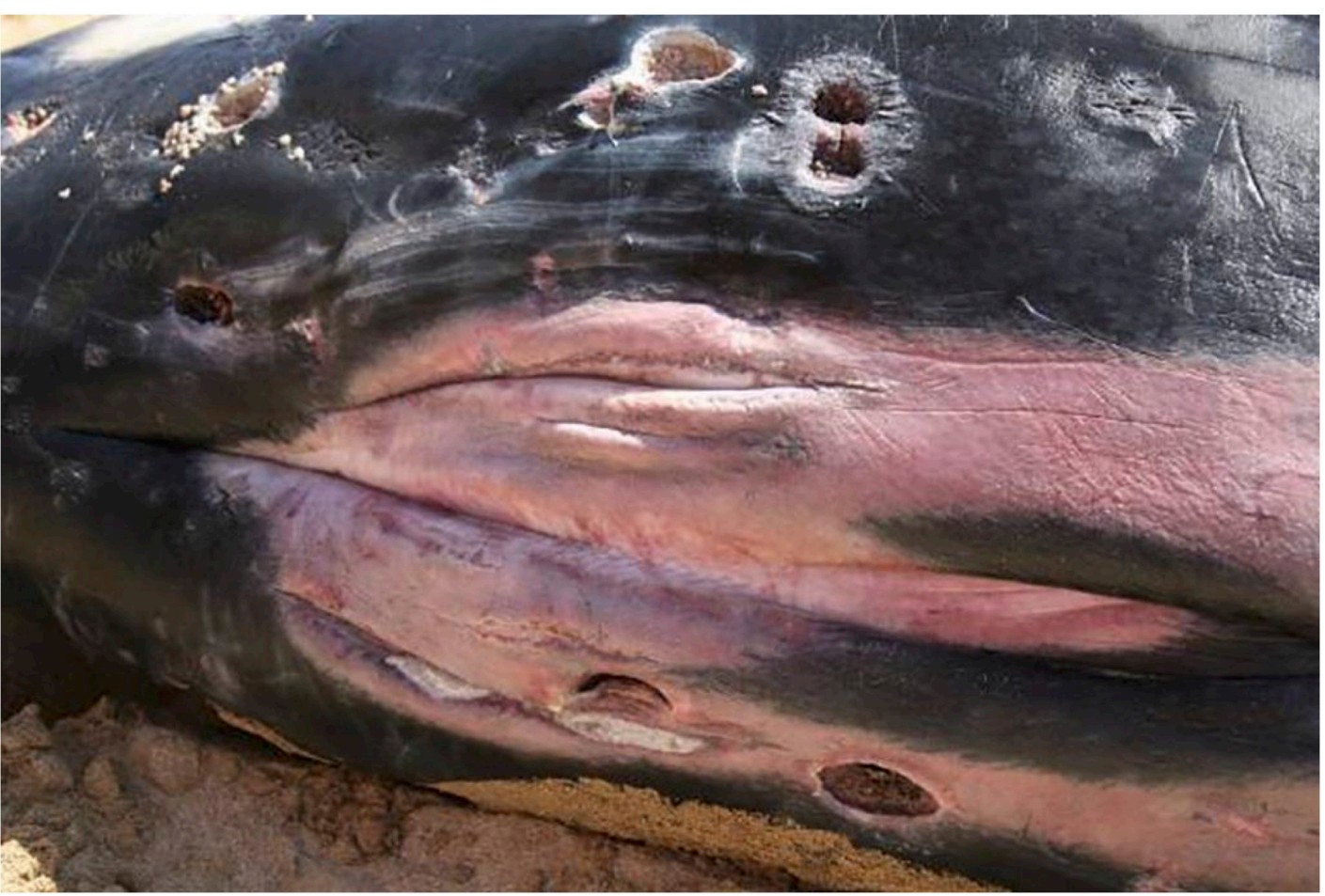

**Fig 5. Irregularly round cutaneous defects often populated by cyamid crustaceans seen on 20081021.** These 3–5 cm diameter lesions are believed to be cookiecutter shark (*Isistius* sp.) bite wounds.

One near term pregnant female and a sub-adult whale were examined, and both were severely emaciated. Based on dehydration and starvation associated with the lack of available prey during this extended extralimital period, nutritional stress was the proximate cause of death.

Two adult animals were diagnosed with emaciation: a female that stranded in Hawaii (ID# 20040406) and a female transient (ID# 20070520) that stranded in British Columbia. Both animals had a variety of incidental findings, but no apparent lesions that may have accounted for the starvation.

### Important ancillary findings

Important ancillary findings were noted in multiple stranded killer whales that had definitive causes of death, as well as in six animals where the cause of death could not be determined (Table 1).

**Cardiovascular.**   Multifocal perivascular and interstitial myocardial fibrosis was observed in two adult females; one from Hawaii (ID# 20040406) and one from British Columbia (ID# 20070520). In both cases, there was no evidence of gross or microscopic sequelae to suggest cardiac insufficiency due to the lesions.

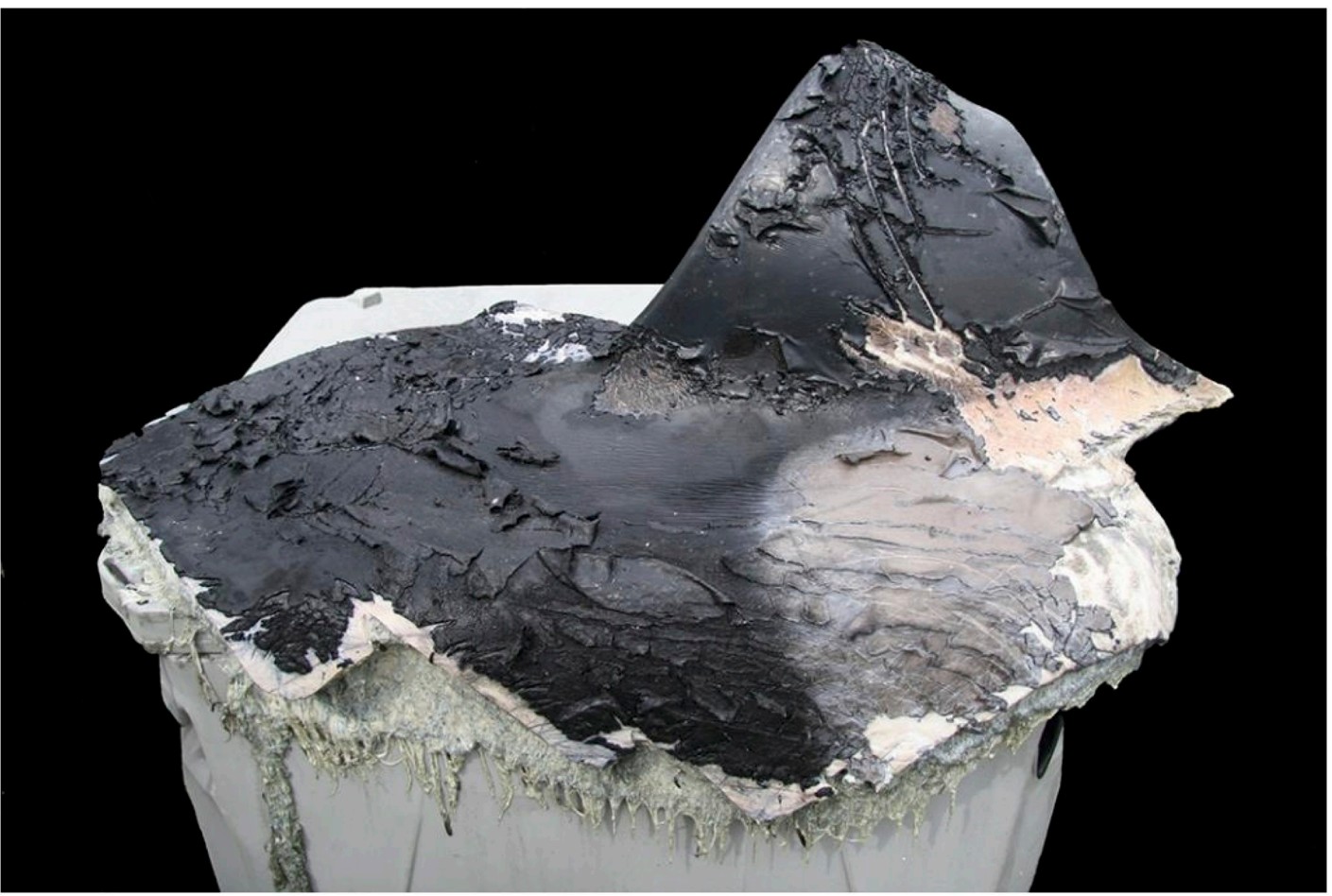

**Fig 6. Skin, blubber and muscle of a transient (ID# T086) that died from a vessel strike.**

**Gastrointestinal.**   Three adult animals had severe tooth wear. Two (ID#s O319 and 20131104a) were definitely, and the third (ID# 20131104b) likely, offshore ecotypes known to specialize in eating sharks [2]. An adult male northern resident killer whale (ID# I046) died of undetermined causes but a barbless treble fish hook was found within the colon. The entire length of the alimentary canal was assessed, with no evidence of gastrointestinal perforation or apparent damage to the mucosa.

**Musculoskeletal.**   Incomplete fusion of the dorsal process of C6 (spina bifida occulta) was identified postmortem in a sub-adult female southern resident (ID# L112) but did not appear to contribute to antemortem morbidity or stranding. The defect was approximately 1cm and the paravertebral soft tissue was unremarkable. Similarly, an adult female transient (ID# T171) presented with severe irregular bone remodeling of the vertebral periosteum involving vertebrae caudal to T6, with no apparent bridging spondylosis. While this change was extensive, it was not believed to have contributed to impaired mobility or the foraging ability of this animal.

**Skin.**   Distinct, round to oval full-thickness lesions consistent with cookiecutter shark (*Isistius* sp.) bite wounds [35] were noted in the skin over the ventral thorax and abdomen of the two killer whales that stranded in Hawaii (ID#s 20040406 and 20081021). In both cases, these wounds had indications of marginal healing and secondary infestation with cyamids,

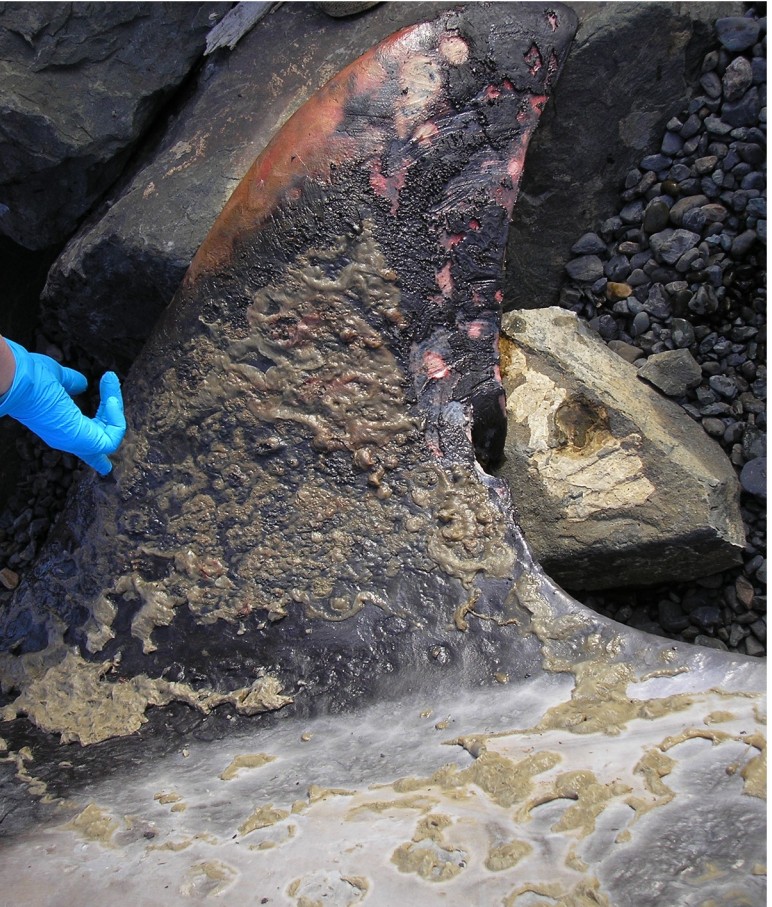

**Fig 7. Skin lesions on dorsal fin of whale (ID# 20111008a) that experienced extended exposure to fresh water.** This moderate to severe, multifocal to diffuse degenerative dermatitis characterized by cutaneous ulcers and erosions, superficial and deep cracking of the skin, and a generalized irregularity to the skin surface was noted in three whales (ID#s 20111008a, 20111008b, and 20111013).

consistent with having occurred prior to stranding and possibly related to malnutrition and debilitation.

The three whales that were in a freshwater river for six weeks (ID#s 20111008a, 20111008b, and 20111013) demonstrated a moderate to severe, multifocal to diffuse erosive and ulcerative dermatitis with superficial and deep epidermal fissures (Fig 7). A mixed population of bacteria and fungi was scattered throughout the ulcerated tissue.

**Reproductive.**   A 154cm southern resident fetus (ID# 20080726) was found with moderate post-mortem decomposition; however, hemorrhage and edema were present within the cervical and thoracic regions suggestive of antemortem trauma, possibly related to dystocia or post-partum injury.

An adult transient female (ID# 20040503) that live stranded and died of an undetermined cause had multiple firm, white, well-defined masses associated with the fallopian tube, uterus and mesovarium (Fig 8). The masses varied in size from 2-8cm in diameter and histologically, were identified as leiomyomas. Immunohistochemistry was positive for vimentin with variable expression of both desmin and smooth muscle actin and staining for CD10 and calmodulin were negative. The contribution of this tumor to impaired fecundity and clinical disease could not be determined. Moderate cholangiohepatitis was also diagnosed in this case.

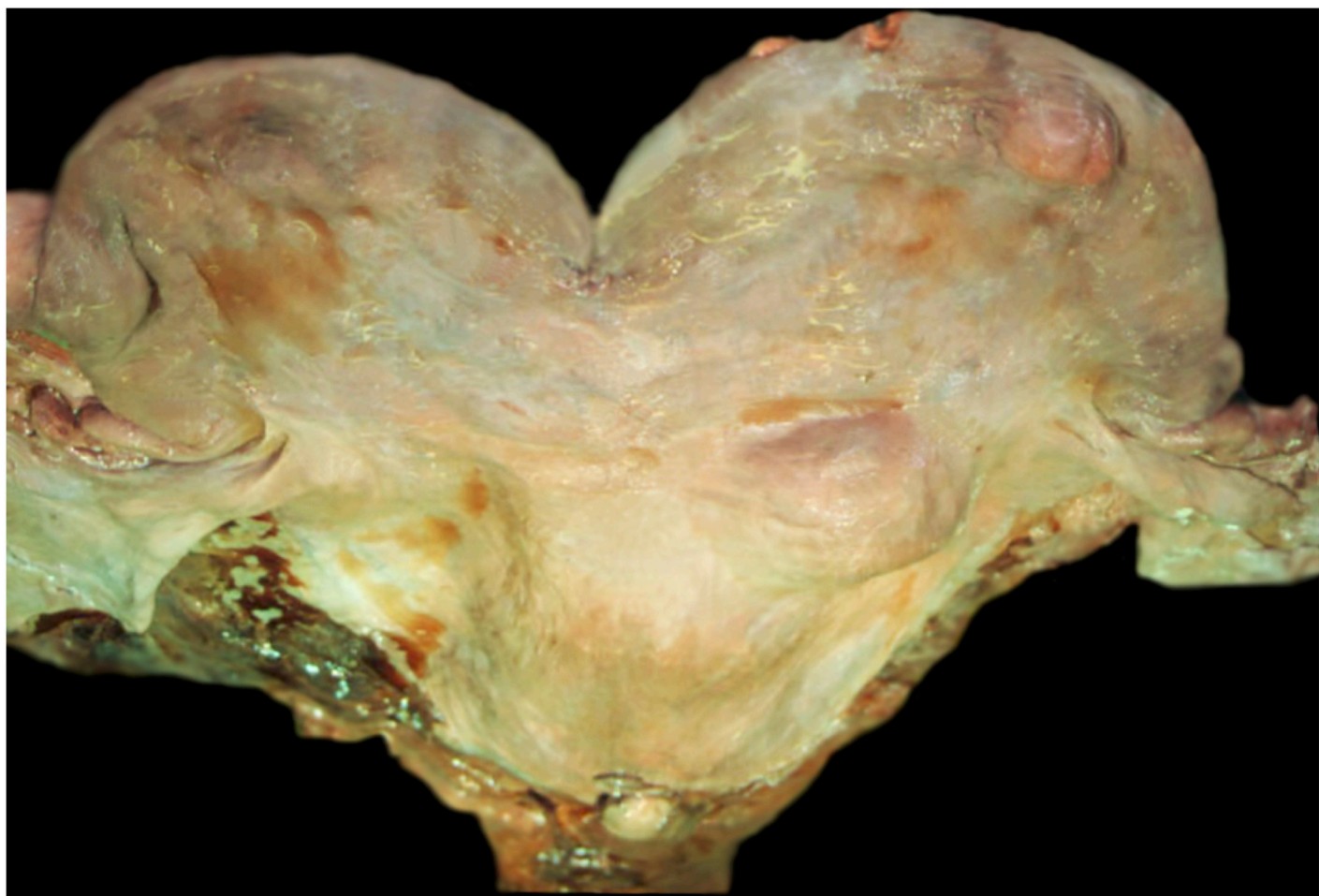

**Fig 8. Multiple leiomyomas associated with the fallopian tube, uterus and mesovarium.** This lesion was identified in adult transient female (ID# 20040503).

**Parasites.** Gastric helminths were observed in five animals. Parasites from three cases (ID#s 20070520, O319, and 20040406) were not speciated, however verminous gastritis was noted in ID# 20070520. Numerous *Anasakis simplex* were observed in the stomach, proximal duodenum and in the lumen of the gall bladder of a nursing 358cm long female Alaskan resident calf (ID# 20050826). The trematode *Odhneriella subtila* was identified in the small intestine of this animal; however, stomach, small intestine and gall bladder were not examined microscopically. A 3-year-old female southern resident (ID# L112) had approximately 30 *Anisakis simplex* aggregated in the nonglandular fore-stomach mucosa. Histopathology of the junction of the glandular and nonglandular compartments revealed hyperplasia of squamous epithelium with ortho- and parakeratotic hyperkeratosis, transverse mucosal clefts and a few superficial, luminal nematode parasites. *Crassicauda* sp. were also detected in the peribullar sinuses and fibrovenous plexus. Sinusitis was observed at necropsy and histopathology showed luminal adult parasites and necrotic exudate contained within edematous mucosa with dilated vascular/sinusoidal spaces demonstrating collapsed profiles of nematodes and very large numbers of thick-shelled nematode eggs containing partially developed larvae. Sloughed sinus epithelium contained moderate numbers of round cell "ghosts" interpreted as infiltrating leukocytes. In one animal (ID# 20110313), a presumptive calcified renal parasitic granuloma was diagnosed microscopically, though nematodes were not identified in the lesion.

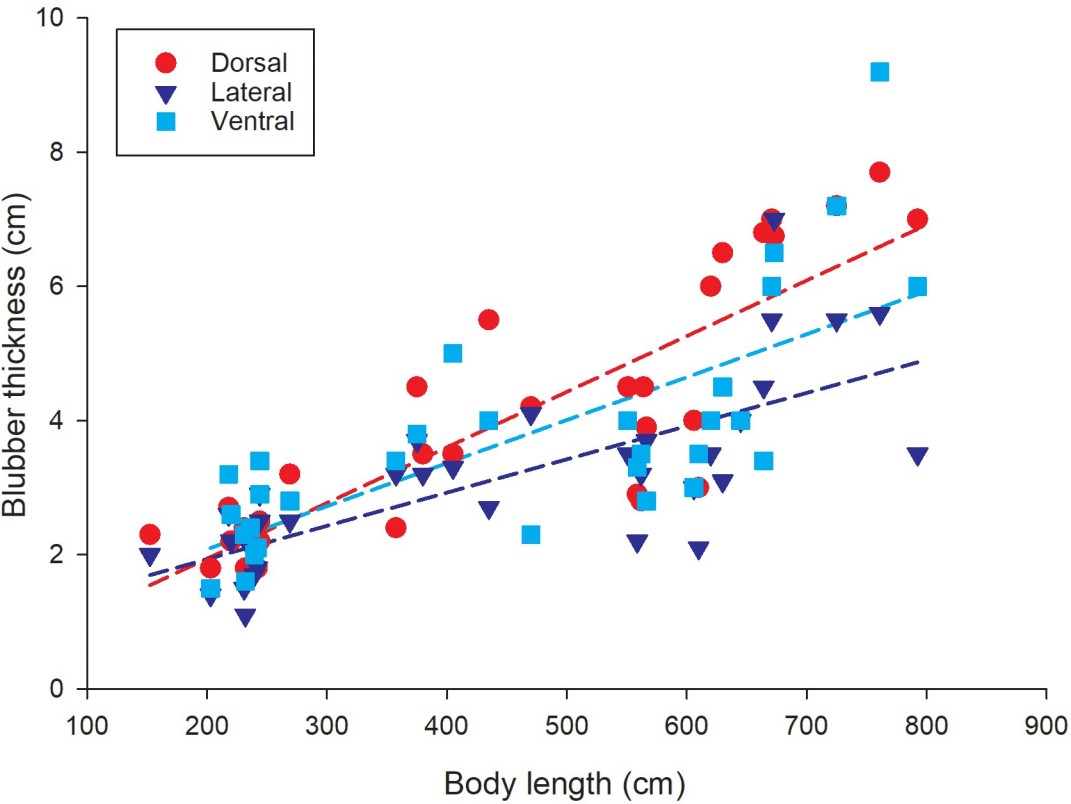

**Fig 9. Blubber thickness in relation to body length.** Blubber thickness measurements at the three sites along the anterior dorsal fin insertion girth measurement line are presented. Least-squares linear regressions are also shown (dashed lines): dorsal blubber thickness (red circles) = 0.3 + 0.008 body length (F = 86.4, $R^2$ = 0.7, $p < 0.001$, n = 33 individuals), lateral blubber thickness (dark blue triangles) = 0.9 + 0.005 body length (F = 36.8, $R^2$ = 0.5, $p < 0.001$, n = 33 individuals), ventral blubber thickness (light blue squares) = 0.8 + 0.006 body length (F = 34.1, $R^2$ = 0.5, $p < 0.001$, n = 31 individuals).

**Body condition index findings.** Blubber thickness ranged from 1.1 to 3.4 cm in the near-term fetus/neonates/calves (152.4 to 357.5 cm body length) and from 2.1 to 9.2 cm in sub-adult/adult killer whales (375 to 792.5 cm body length). The generalized linear model found a significant association between body length and blubber depth at all three sites for all animals combined (p<0.001, for all sites; Fig 9) as well as for the sub-adult/adult killer whales (p<0.05, all sites). However, there is no association between body length and blubber depth at any of three sites (dorsal: p = 0.4, lateral: p = 0.06, ventral: p = 0.09) for the fetus/neonates/calves. Based on the results of the GLM, an ANCOVA and an ANOVA were used to assess variability in blubber thickness across the three sites for the sub-adults/adults and fetus/neonates/calves, respectively. For the older animals, blubber thickness increases with body length (p < 0.001), there is no interaction between blubber sample site and body length, and blubber thickness differs significantly across measurement sites (P = 0.01). Pairwise comparisons show that dorsal blubber thickness is greater than lateral blubber thickness (p = 0.008). In contrast, for the fetus/neonates/calves, blubber thickness does not differ across the three blubber measurement sites (P = 0.2).

There is no difference in body length between killer whales that died from infectious or nutritional causes and killer whales that incurred trauma (Mann-Whitney Rank Sum test, p = 0.2). As expected, blubber thicknesses for individuals (all ages combined) that died from

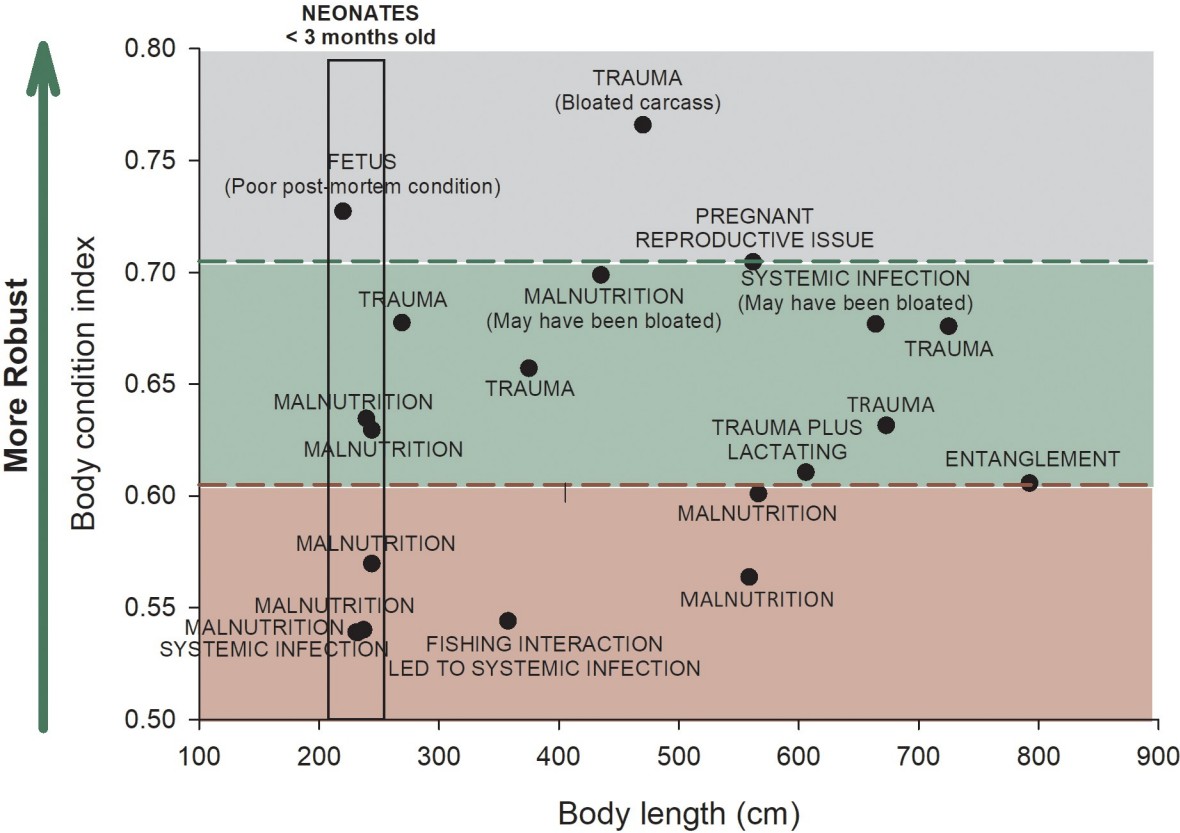

**Fig 10. Relationship between body condition index, body length, and cause of mortality.** BCI for killer whales with known causes of death are presented. Poor BCI (BCI, girth at the anterior insertion of the dorsal fin/body length) values, good BCI values, and artifactually inflated BCI values are designated by the red, green, and gray bands, respectively. The minimum and maximum values for the "good BCI" range (green band) are based on the minimum and 75% BCI values for animals that died from trauma. The BCI value for the pregnant female was also nearly identical to the 75% BCI value for animals that died from trauma. The animal with the greatest BCI (0.77) also died from trauma, but the necropsy report specifically states that the carcass was bloated (gray band). BCI values less than the minimum value for animals that died from trauma are considered to be in poor condition (red band). There is no relationship between BCI and body length.

trauma are greater than those for individuals that died from systemic infection or malnutrition (dorsal: Mann-Whitney Rank Sum test, p = 0.03; lateral: one-tailed t-test p = 0.01, ventral: one-tailed t-test, p = 0.02).

In contrast to blubber thickness, there is no relationship between body length and the body condition index (BCI; Spearman Rank Order Correlation, p = 0.3). BCI ranged from 0.54 to 0.73 for the near-term fetus and calves and 0.56 to 0.77 for older animals (Fig 10). Instead, BCI values appear to be related to cause of death (Fig 10). BCIs for individuals that died from trauma (range = 0.61–0.77, mean = 0.67 ± 0.05 SD) are significantly greater (Mann-Whitney Rank Sum test, p = 0.04) than BCIs for individuals that died from infectious and nutritional causes (range = 0.54–0.70, mean = 0.59 ± 0.06 SD). Four out of seven (57%) animals that died from nutritional causes had BCI values that were ≤ 0.60 and three out of four (75%) animals that died from systemic infections had BCI values that were ≤ 0.56. All six killer whales that died from trauma and the one killer whale that died from entanglement had robust BCI values (> 0.60), which would be expected for healthy animals that died peracutely from a catastrophic event. BCI values ≥ 0.70 may be artifactually inflated due to carcass bloat or pregnancy (Fig 10).

## Discussion

This study demonstrates the benefits of necropsy exams to better understand killer whale health. Of 53 cases, cause of death was determined for 22 (42%) and nine additional animals demonstrated secondary findings of significant importance for population health. These include cases of human interaction, such as hooks and vessel trauma, as well as congenital abnormalities, abortion and nutritional status. Collectively, they show important return on investment for conducting thorough postmortem examinations.

### Nutrition

Determination of the nutritional condition of stranded animals in the context of pathologic findings in this study was a challenge. Cetaceans have minimal visceral adipose stores. The greatest reserves accumulate in the subcutaneous adipose (blubber) layer and vary with food availability, elements of seasonality including hormones cycles, reproductive status, water temperatures, migration and prey availability [36–38]. Blubber loss also varies by site in emaciated cetaceans. For example, blubber thickness in the thorax of starved porpoises (9-11mm) was reduced to 50–60% of that of normal animals (18-20mm); however, very little tailstock blubber was lost [36]. Although blubber thickness measurements can inform nutritional condition assessments, the interpretation of these measurements can be complicated.

There are many potential underlying causes of malnutrition and poor body condition. These include reduced prey availability; impaired ability to successfully forage, apprehend, or consume prey (such as with a broken or deformed jaw); inappetence due to an underlying condition; reduced intestinal absorption of nutrients (malabsorption); increased protein loss such as with protein-losing nephropathy or enteropathy; or increased metabolic demands such as is seen in pregnancy, lactation, increased energy expenditure to meet physiologic needs, or with cachexia of chronic inflammation. Because the necropsy examination is a "health snapshot" at the time of death, the role of weight loss as a precursor versus the product of general poor health can be difficult to determine. Typically, a pathogenesis of cause may be inferred based on staging significant necropsy findings and review of environmental information.

Observer variation in determination of the nutritional status of animals at the time of death further complicated this assessment. Based on review of field observations and necropsy reports, relatively few animals were identified as thin or emaciated. However, in several cases, field observations indicating that animals were in good condition were uncorroborated by blubber thickness (relative to body length) and/or BCI values. While blubber depth was the prime nutritional condition objective measure for this determination in dead animals, site- and body length- specific ranges had not yet been produced to make these measurements more than a crude evaluation. In terrestrial animals and humans, serous atrophy of fat and bone marrow are key indicators of emaciation. Due to the limited distribution and absolute amount of bone marrow in cetaceans, this method has not been routinely employed to assess nutritional status in stranded killer whales. Conventional indicators of reduced or poor body condition in cetaceans include a loss of fat in the nuchal crest resulting in formation of a cervical indentation, a condition known as "peanut head." Additional subjective determinations include reduction in blubber identified by concavity along the epaxials, a visible rib pattern in the skin over the thorax, loss of pericardial and epicardial fat, and generalized muscle loss. These can be difficult to detect in free-swimming animals and are likely late-term indicators of poor body condition. In fact, 11 of 13 southern resident killer whales noted with these signs between 1994 and 2008 subsequently died [39].

One clear conclusion from this review is that a defined body condition score with objective measures is needed to better assess killer whale nutritional status. In this case series, blubber

thickness increased with body length and as few animals of similar body length strand, it is difficult to rely on blubber thickness alone as an indicator of nutritional condition. Furthermore, in terrestrial mammals, emaciation is characterized by a loss of subcutaneous, epicardial, and mesenteric adipose as well as an associated prominence of the ribs, vertebrae and pelvic bones. Additional changes can include muscle, liver and fat atrophy. The body condition index, which accounts for changes in total body girth relative to body length, correlated with COD in most of the stranded killer whales. As subtle changes in blubber thickness can occur, BCI provides a more sensitive assessment of body condition. For example, a lactating southern resident female that stranded in 2002 (ID# L060) and an entangled male transient killer whale that stranded in 2015 (ID# 20150418) had the lowest BCIs of the more robust animals. Lactation is a metabolically expensive process [40] that can deplete energy stores [41], and the entangled killer whale died with fishing gear still attached, suggesting the animal was trailing gear for some time prior to death. Decreased body condition is often noted in entangled baleen whales and is related to increased drag caused by entangling gear and subsequent increased energy expenditure required for swimming [42].

Because blubber reduction may not be as significant as changes in epaxial and hypaxial musculature beneath the blubber layer, BCI values are likely better indicators of body condition than blubber thickness measurements alone. For example, a recently stranded 300 cm long killer whale calf (ID# 20180530) that died of malnutrition had blubber thickness measurements not consistent with emaciation; however, its BCI was 0.57, placing it within the range of a malnourished animal (Fig 10). In this case, blubber measurements alone would not have properly assessed the body condition in this animal. In other odontocetes, the limitations of interpreting blubber measurements to overall nutritional status of individuals have been attributed to a lack of relationship between blubber thickness and blubber lipid content [43] and variability of blubber depth across the body.

However, the post mortem state of a carcass should be considered when assessing BCI. Two killer whales, one that died from malnutrition (ID# 20151015) and one that died from a systemic infection (ID# L095) had relatively high BCIs but presented with advanced autolysis, which contributed to bloat that was not noted on necropsy. Based on morphometric data for the three killer whales with body condition indices of > 0.70 (Fig 10), bloat, poor post-mortem condition with fat lysis, and pregnancy can artifactually inflate the body condition index. This suggests that using BCI as an indicator of health status of stranded killer whales is best limited to carcasses in moderate to good post-mortem condition. Other criteria, such as blubber lipid content, gross appearance of the blubber (oily vs. dry), and muscle condition may be used for killer whales found in advanced autolysis.

Application of new technologies in the health assessment of free ranging killer whales can provide a moving picture of animal nutritional condition. For example, remote drone photogrammetry can document individual animal and population-level condition over time. Although they do not provide an indication of the underlying cause for poor nutritional condition, measurements of free-swimming whales collected by drone aircraft offer promise for monitoring animal's body condition [44]. A post-mortem analog for these measurements and scoring of fat reserves in carcasses should be validated so that additional insights may be garnered from post mortem case material.

Differentiating causes of poor body condition (e.g. infection versus nutritional) in killer whales solely using BCI values for stranded individuals or aerial photogrammetric images for free-swimming whales, is impossible, given analytical limitations, including sample size. Despite some mischaracterizations of observed body condition, there was still a preponderance of more robust animals in the stranding dataset. For example, approximately two thirds of the stranded killer whales for which valid BCIs were determined were in good condition. This

suggests that animals in moderate to good body condition are positively buoyant and are more likely to be recovered than animals in poor condition with less blubber. However, since one third of these killer whales had poor BCI values, it is evident that other factors (e.g. carcass bloat, geographic location) beyond body condition contribute to the likelihood of carcass recovery, as has been described [45]. Furthermore, as mentioned above, biological samples are necessary to tease apart mechanisms leading to poor body condition.

## Human interaction: Trauma and fishing

Human interaction cases were identified in all age classes (1 calf—ID# 20050826; two sub-adults—ID#s L098 and C021; and two adults—ID#s T086 and I046). Identification of vessel strike-related trauma demonstrates that human interaction is a significant cause of morbidity or mortality in killer whales. These findings suggest that vessel strikes may be an important threat, particularly in the endangered southern resident killer whale population that frequents areas near large human populations and shipping lanes. Indeed, as described above, southern resident killer whale J034 died from trauma consistent with a vessel strike.

Traumatic injuries distinct to vessel strike were identified in six strandings. In three cases (ID#s O319, N018, and L112) the origin of trauma could not be determined. Conspecific aggression was a prime consideration for offshore O319. This animal presented with a simple complete transverse closed rib fracture not commonly associated with catastrophic blunt force injury, but rather with a focused traumatic event, such as intraspecies aggression associated with a forceful bite or head strike. The transient N018 also had a focal septic comminuted closed vertebral fracture, which may also represent intraspecific aggression. In contrast, the sub-adult southern resident L112 exhibited massive blunt force trauma and lesions consistent with a glancing blow. While the origin of the trauma could not be determined, a vessel strike could not be ruled out. Based on field observations or necropsy findings, vessel strike was confirmed in two animals (ID#s L098 and T086) and suspect for one additional whale (ID# C021) (9.3% of trauma cases were related to vessel strike). Moreover, an 18-year old male southern resident (ID# J034) was found dead near Sechelt, B.C. (December 20th, 2016) with extensive subcutaneous and epaxial hematoma in the left dorsolateral thorax. The hemorrhage indicated that J034 was alive at the time of impact and based on the extent and severity of the lesion, was likely struck by a vessel. In a separate case series, Williams and O'Hara [46] identified ten killer whales that were struck by vessels in British Columbia waters between 1995 and 2005. Five were non-fatal strikes, two were fatal (one being L098), two were classified as serious injury and one animal died a year after being injured. Prior to that, Ford et al. [47] identified two northern residents that were struck by propellers and survived and described a killer whale calf (suspected to be A021) that was struck by a ferry in 1973 and likely later died. Visser [48] identified propeller scars on two of 117 photo-identified killer whales in New Zealand and reported an additional fatal vessel strike in a third animal. Historically, vessel strike has not been considered an important anthropogenic cause of morbidity or mortality in killer whales; however, based on findings from this pathology review and other observations of vessel strike [46, 47], this risk factor may be an underappreciated but important threat to the population status of endangered killer whales in the eastern Pacific.

Although fatal fishing interactions are infrequent in stranded animals, ingestion of fish hooks may pose a health concern, particularly in younger, less experienced animals. Fishing hooks have been documented in stranded resident killer whales as early as 1973. Ford et al. [49] reported that of 8 examined resident killer whales, two contained hooks or lures designed for salmon fishing and two other animals had ingested hooks used for Pacific halibut fishing. In Alaska, killer whales are known to depredate sablefish (*Anoplopoma fimbria*), Pacific halibut

and Greenland turbot (*Reinhardtius hippoglossoides*) longlines [17]. Depredation by killer whales has also been documented in other regions of the world. Off the coast of the Crozet Islands, reproductive success of killer whales has been associated with consumption of Patagonian toothfish (*Dissotichus elegenoides*) off longlines in contrast to conspecifics that do not rely on hooked bait fish [50].

Although no southern resident killer whales have been reported to have died from fishery interactions to date, based on prior field observations and case findings in other regions in the eastern Pacific, southern residents do interact with fisheries, but incidents may not be well documented. On August 1, 2015 a 12-year-old male southern resident killer whale (ID# J039) was photographed with a salmon flasher hanging from its mouth. The rigging was consistent with a squid lure 3/4" barbless hook (point to shank) on 30lb test line and if ingested, the hook would have extended up to 24" to 44" into his esophagus. This whale was monitored and the flasher was present 5 days later, then lost the following day. Currently J039 is alive with no apparent adverse effects from this incident. Evidence of fishing interaction appears more frequently with younger animals and suggests that depredating fish from longlines without ingesting the hook could be a learned behavior that improves with experience. The calf (ID# 20050826) that died of septicemia, secondary to hook ingestion, may have been in the process of learning this behavior. Of course, it is possible for whales to capture and consume fish that have broken away from fishing gear and taken some or all of the gear with them. Alternately, an older animal that depredated the fish and hook could have shared the depredated fish with this younger animal [51]. Due to the small sample size, the potential impact of depredation on resident whale populations could not be determined. The incidence of depredation may vary by fishing intensity in the region, cultural learning in the pod, hook type and possibly age structure of the whale population. Based on these findings, fisheries interactions should be further evaluated as a potential health hazard to killer whales and appropriate practices considered to limit these events if they are found to jeopardize population viability.

## Infectious disease

Infectious disease was identified as a proximate cause of death in two whales (ID#s 20051207 and 20070520) and an ultimate cause of death secondary to trauma in two additional animals (ID#s 20050826 and N018). Bacterial infections also were associated with ingestion of foreign bodies (septicemia secondary to a fishing hook ID# 20050826) and were likely involved with the fibrinous peritonitis diagnosed in an emaciated female adult killer whale (ID# 20070520). The mixed bacterial flora recovered from sampled tissues in both cases suggest disruption of the gastrointestinal mucosa as the source of bacterial invasion and infection. Only one case of bacterial septicemia from a single agent (*Salmonella* Newport) was recovered. In this case, the origin of the infection was likely maternal bacteremia with localization to the placenta and introduction to the fetus via the umbilicus [8]. The proliferative costochondritis (ribs) and age of this animal indicated fetal sepsis with possible growth plate involvement. *Salmonella* sp. was also isolated from an adult female killer whale (ID# 20081021); however, in this case, the lack of associated gross and histopathologic lesions suggest that this animal was an asymptomatic carrier, rather than actively infected. In examining future cases where *Salmonella* spp. may be recovered, it is imperative that the isolate is placed in appropriate clinical context with regards to associated nutritional status and pathologic findings of the animal. Serotyping or genotyping should be pursued to determine if the bacteria may have been terrestrially sourced or may cycle exclusively in the marine environment.

Despite testing of postmortem serum, screening tissues by molecular studies and looking for cytoplasmic inclusions and syncytia microscopically, morbilliviruses, such as cetacean

morbillivirus and morbillivirus antibodies have not been detected. This further supports the potential that exposure to these viruses could increase extinction risk in immunologically naïve and endangered killer whale populations like the southern residents [10, 52].

## Reproductive disease

Fecal hormone analysis of southern resident killer whales [53] documented early embryonic loss and fetal resorption or abortions in 69% of detectable pregnancies with approximately 20% late term pregnancy losses. These findings coupled with up to 50% of neonate and yearling calf mortalities (in animals sampled) substantiate the importance of reproductive disease in southern resident killer whales. Prey availability and contaminant loads were prime considerations in reproductive failure in this study and in our case series only a single abortion was documented (ID# 20080726). However, reproductive failure was also confirmed in a 19-year-old full term pregnant female southern resident killer whale (ID# J032) that stranded outside of our time series in 2014, due to fetal loss and septicemia. Although the cause of abortion and *in utero* mortality are not readily apparent, the scat analysis field study [53] and cases in this post mortem cohort highlight that reproductive disease does occur in this species and contributing factors such as nutritional status, pathogen exposure and anthropogenic activities should be investigated.

## Cardiovascular pathology

Myocardial fibrosis was diagnosed in two adult killer whales (ID#s 20040406 and 20070520). This lesion is commonly identified in older odontocetes and seems to correlate with increasing age. The pathogenesis of this lesion is likely complex and potential causes include toxin exposure, such as domoic acid or mercury, resolving and possibly progressive fibrosis, inflammation, pathogens, nutritional (vitamin E or selenium deficiency), stress reactions, gas bubble disease, and serotonin excess [54, 55]. At present, the etiology of this condition is unknown. The lack of associated cardiomyopathy and pulmonary and hepatic congestion discounts impaired cardiovascular function as the COD; however, the lack of these lesions does not preclude the possibility of an arrythmia or other non-congestive cardiac abnormalities.

## Congenital disease

Congenital disorders were identified as the proximate cause of death in a neonate and calf. These conditions were incompatible with neonatal development beyond a few weeks or months post-partum. In contrast, a variety of skeletal malformations have been reported in other killer whale populations and have been identified in all age classes. Both thoracic kyphosis and scoliosis have been documented in Norwegian killer whales [56], and one case involved a calf with no evidence of trauma. In a second calf (neonate) with scoliosis caudal to the dorsal fin, there was observed trauma to the dorsal fin suggesting that this spinal pathology may have been acquired. Lordosis and kyphosis were described in two live New Zealand killer whales as well [56].

While no vertebral malformations resulting in spinal column deviations were identified in this study set, a single case of spina bifida occulta characterized by a 1cm defect in the dorsal process of C6 was identified (ID# L112) [57]. This vertebral change has been seen by one of the authors (JS) in killer whale skeletons retained in museum collections. It may be a common defect with no apparent clinical or pathologic consequence.

Hiatal hernia has not been previously been recognized in cetaceans. Familial inheritance of hiatal hernia occurs in humans with evidence of direct male-to-male transmission and suggests an autosomal dominant mode of inheritance [58]. Because of inbreeding concerns with certain

males being over-represented as sires in the southern resident killer whale population, heritable conditions like this could represent population threats [59]. In this case, increased intra-abdominal pressure associated with dystocia and protrusion of the stomach through the hiatus may also be a consideration.

Three full term neonates presented in suboptimal body condition (ID#s 20120326, 20130904 and 20100505) and based on the stage of development and BCI, *in utero* malnutrition was a prime consideration. In one neonate, no ingesta was noted in the stomach. These cases may be related to maternal malnutrition in the latter stages of pregnancy with possible post-partum maternal separation, loss, neglect or a nonviable neonate.

## Environmental events

Environmental conditions can be sufficient to result in the death of killer whales. Two transient killer whales (ID#s 20060728a and b) that stranded near Cordova, Alaska were likely feeding in shallow flats when the ebbing tide receded. This area is known for its long shallow intertidal zone characterized by rapidly drops in water levels. A series of fairly extreme tidal cycles occurred in the area over a three-day period when the stranding occurred and is likely that these animals moved too far inshore at high tide and stranded when the tide dropped. A similar case occurred in Tofino, British Columbia (Canada) in 1976 when an adult male transient stranded on a large tidal flat, was unable to re-float and died [60]. In 2011, a male and female transient stranded in Marcus Passage, British Columbia while swimming over sand flats during an ebb tide, however these two animals refloated themselves later and swam off [61].

The second environmental event occurred in 2011 when three whales (ID#s 20111008a and b and 20111013) swam up to 50km into the Nushagak River from the Bristol Bay near the town of Dillingham, Alaska. All three whales demonstrated degenerative changes in their skin with florid secondary bacterial and fungal colonization and overgrowth.

## Parasites

Complete necropsies provide an opportunity to address knowledge gaps about the composition and pathology associated with metazoan and protozoal parasites in killer whales. Metazoan parasites previously reported in killer whales include: the Acanthocephalans *Bolbosoma nipponicum* and *B. physeteris*, the cestodes *Diphyllobothrium physeteris* and *D. polyrugosum* [62, 63], the nematodes *Anisakis pacificus*, *A. simplex* [62–66], and *Halocercus* sp. [67], and the trematodes *Campula* sp., *Fasciola skrjabini*, *Hadwenius subtilla*, *Oschmarinella albamarina*, *Phyllobothrium* sp., and *Trigonocotyle spasskyi* [62–64, 68]. Although *Anasakis* sp. have been reported in killer whales, ID# 20070520 may be the first case with associated gastritis. The trematode *Odhneriella subtila* identified in the small intestine of ID# 20050826 was not unusual and may be the same species previously reported in a killer whale described as *Hadwenius subtila* by Adams and Rausch [69]. The identification of *Crassicauda* sp. in the peribullar space of L112 likely represents a novel nematode reported in killer whales. Interestingly, *Crassicauda* sp. were also found in the peri-bullar region of an 18-year-old female southern resident (ID# J032; Table 2). Sinusitis was grossly (for L112) as well as microscopically (for J032) associated with the presence of these nematodes.

*Toxoplasma gondii* antibodies have been detected in a killer whale from Japan [70, 71], and we detected dual infection with *T. gondii* and *Sarcocystis neurona* in an adult female (ID# N018) [72]. The clinical and pathologic consequences of protozoal parasites on free-ranging killer whales is not known. However, these parasites are terrestrially sourced and efforts to mitigate discharge of infectious pathogens by appropriate control measures is critical, particularly as the largest water borne infection of *T. gondii* in humans worldwide was reported in municipal

drinking water in Victoria, British Columbia [73]. We did not detect the ciliate, *Kyaroikeus cetarius*, which has been identified in 5 of 6 captive killer whales [74], in any stranded animals.

This case series sets the foundation for understanding killer whale mortality and health concerns and provides a basis for enhanced monitoring and improved management of killer whales, especially for endangered populations. Consistency in sampling and morphometric measurement protocols is critical for continuity of health monitoring in killer whales. Because blubber thickness varies by body length, BCI may be a more suitable tool for assessing body condition in deceased killer whales. However, the finding that neither blubber thickness nor BCI measurements are adequate for differentiating proximate mechanisms for poor body condition has implications for monitoring health in free-ranging killer whales. Without additional biological samples, it is impossible to differentiate animals in poor body condition due to metabolic imbalance, such as during periods of reduced prey availability, from those that are diseased. The results of this case series also suggest that human interaction cases may be more prevalent than previously understood for killer whales in the northeastern Pacific. Continued analysis of necropsy cases will be important for adaptive management efforts.

## Acknowledgments

Samples were collected and evaluated under multiple Stranding Agreements, the US Marine Mammal Protection Act, Canadian Fisheries and Species at Risk Acts, and licenses to hold marine mammal tissues provided to responders, field biologists and veterinarians. Specific acknowledgements go out to logistics and field response personnel and volunteers from regional stranding networks coordinated through NOAA and Fisheries and Oceans Canada, as well as regional stranding coordinators and veterinarians throughout the western seaboard of North America and the Hawaiian Islands, especially A. Jensen, S. Johnson, J. Olson, K. Savage, A. Traxler, and K. West. We also thank Federal Fisheries Officers, State Wildlife Enforcement Officers, and US Fish and Wildlife Inspectors who provided invaluable assistance with collection and transport of samples. We specifically thank the staff at the numerous reference laboratories used, including the University of Illinois, ZAPP, UC Davis Wildlife Health Center, NOAA Northwest Fisheries Science Center (especially G. Ylitalo and K. Parsons). We thank G. Anchor, University of Illinois, for running the Generalized Linear Model analysis. In-kind support was provided by Fisheries and Oceans, Canada, the Vancouver Aquarium Research Program, the SeaDoc Society (especially E. Nilson and E. Ashley), SeaWorld (especially A. Mena), the Animal Health Center of the BC Ministry of Agriculture, the Washington Department of Fish and Wildlife, numerous First Nations, Alaska Native, and Inuit Communities, and other organizations. We thank the private citizens and whale watch operators for ongoing surveillance and reporting dead animals and the numerous regional stranding networks and organizations (including the Whale Museum, Vancouver Aquarium, and Strawberry Isle Marine Research Society) for helping collect samples. We are grateful to L. O'Keefe for creating Fig 1. Dr. M. Haulena of the Vancouver Aquarium provided an early review of the manuscript that strengthened it. Samples have been archived at the Burke Museum, California Academy of Sciences, Royal British Columbia Museum, Ontario Museum of Natural History, University of British Columbia, Fisheries and Oceans Canada, and Seattle Aquarium with the caveat that case material is available upon request for future investigations to better understand the natural history of these animals.

## Author Contributions

**Conceptualization:** Stephen Raverty, Judy St. Leger, Dawn P. Noren, David S. Rotstein, Frances M. D. Gulland, Teri Rowles, Lynne Barre, Graeme Ellis, Joseph K. Gaydos.

**Data curation:** Stephen Raverty, Joseph K. Gaydos.

**Formal analysis:** Stephen Raverty, Judy St. Leger, Dawn P. Noren, Kathy Burek Huntington, David S. Rotstein, Joseph K. Gaydos.

**Funding acquisition:** Stephen Raverty, Judy St. Leger, Teri Rowles, Lynne Barre, Paul Cottrell, Joseph K. Gaydos.

**Investigation:** Stephen Raverty, Judy St. Leger, Dawn P. Noren, Kathy Burek Huntington, John K. B. Ford, M. Bradley Hanson, Dyanna M. Lambourn, Jessie Huggins, Martha A. Delaney, Lisa Spaven, Paul Cottrell, Tracey Goldstein, Karen Terio, Debbie Duffield, Jim Rice, Joseph K. Gaydos.

**Methodology:** Stephen Raverty, Judy St. Leger, Dawn P. Noren, Joseph K. Gaydos.

**Project administration:** Stephen Raverty, Judy St. Leger, Dawn P. Noren, Joseph K. Gaydos.

**Resources:** Stephen Raverty, Judy St. Leger, Kathy Burek Huntington, Frances M. D. Gulland, John K. B. Ford, M. Bradley Hanson, Dyanna M. Lambourn, Jessie Huggins, Martha A. Delaney, Lisa Spaven, Paul Cottrell, Graeme Ellis, Tracey Goldstein, Karen Terio, Debbie Duffield, Jim Rice, Joseph K. Gaydos.

**Supervision:** Stephen Raverty, Judy St. Leger, Joseph K. Gaydos.

**Visualization:** Stephen Raverty, Judy St. Leger, Dawn P. Noren, Joseph K. Gaydos.

**Writing – original draft:** Stephen Raverty, Judy St. Leger, Dawn P. Noren, Joseph K. Gaydos.

**Writing – review & editing:** Stephen Raverty, Judy St. Leger, Dawn P. Noren, Kathy Burek Huntington, David S. Rotstein, Frances M. D. Gulland, John K. B. Ford, M. Bradley Hanson, Dyanna M. Lambourn, Jessie Huggins, Martha A. Delaney, Lisa Spaven, Teri Rowles, Lynne Barre, Paul Cottrell, Graeme Ellis, Tracey Goldstein, Karen Terio, Debbie Duffield, Jim Rice, Joseph K. Gaydos.

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
