## [Decision Letter · Decision Letter 0]

16 Sep 2020

PONE-D-20-23093

Pathology findings and correlation with body condition index in stranded killer whales (Orcinus orca) in the northeastern Pacific and Hawaii from 2004 and 2013

PLOS ONE

Dear Dr. Gaydos,

Thank you for submitting your manuscript to PLOS ONE. After careful consideration, we feel that it has merit but does not fully meet PLOS ONE’s publication criteria as it currently stands. Therefore, we invite you to submit a revised version of the manuscript that addresses the points raised during the review process.

Both reviewers were very positive regarding this manuscript and its results and they both agree this is an important contribution to the field and will help many researchers working in this and similar species.  I think it is important that the authors re-evaluate the statistics and data analyses they included, since one of the reviewers points out that the statistical tests used for the data may not fit due to criteria regarding data normality, etc.  For that reason I suggest that the authors double check this comments and consider alternative ways to analyze their data.  

We look forward to receiving your revised manuscript.

Kind regards,

Susana Caballero, PhD

Academic Editor

PLOS ONE

Journal Requirements:

Additional Editor Comments (if provided):

Both reviewers were very positive regarding this manuscript and its results and they both agree this is an important contribution to the field and will help many researchers working in this and similar species. I think it is important that the authors re-evaluate the statistics and data analyses they included, since one of the reviewers points out that the statistical tests used for the data may not fit due to criteria regarding data normality, etc. For that reason I suggest that the authors double check this comments and consider alternative ways to analyze their data.

Reviewers' comments:

Reviewer's Responses to Questions

**Comments to the Author**

1. Is the manuscript technically sound, and do the data support the conclusions?

Reviewer #1: Yes

Reviewer #2: Yes

2. Has the statistical analysis been performed appropriately and rigorously? 

Reviewer #1: Yes

Reviewer #2: No

3. Have the authors made all data underlying the findings in their manuscript fully available?

Reviewer #1: Yes

Reviewer #2: Yes

4. Is the manuscript presented in an intelligible fashion and written in standard English?

Reviewer #1: Yes

Reviewer #2: Yes

5. Review Comments to the Author

Reviewer #1: I congratulate the authors on this piece of work. I found it interesting and easy to read, and a valuable addition to existing knowledge. I have only minor comments to make, mostly of an editorial rather than a content nature. These are detailed below.

Line 162 (BCI equation) - 'enterior' should be 'anterior'

Line 190-191 - this should either be 'association .... was evaluated' or 'associations.... were evaluated'

Line 231 - add 'which' ("was an offshore calf which had a hiatal hernia...")

Line 258 - should be bone remodelling rather than boney

Line 392 - add 'with' or 'in' (was found with moderate...)

Line 395. Meconium staining of what? Also please detail the signs of fetal distress that you screened for but were not present

Line 677 - Please remove the comma after peritonitis

LIne 678 - Bacteria should be bacterial

Line 694 - I think this should be cytpolasmic rather than cytoplastic?

Line 716 - 'concerns' is not a particularly scientific word. Is there a more precise subheading that could be used here?

Line 718 and 719 - 'this' lesion is used in the first sentence and then 'these lesions' in the following one. Please be consistent

Line 719 - 722. Please include references for pathogenesis of myocardial fibrosis. The sentence that begins with "The pathogenesis" requires some work. What do you mean by 'long past inflammation'. The word 'infectious' can be removed (pathogens is sufficient). 'Diving issues' is not a scientific term; please use more precise language here. Finally, the absence of cardiomyopathy, pulmonary congestion and hepatic congestion does not rule out the possibilty of non-congestive cardiac abnormalities such as arrhythmias, as has been described in association with myocardial fibrosis in humans and domestic species.

A final general comment. I found the conclusions about blubber depth and BCI particularly interesting and relevant to researchers working on cetacean health and disease, and wondered if it would be worth re-emphasising these findings in the concluding paragraph.

Reviewer #2: This study of causes of death of killer whales is an important addition to the marine mammal literature. However, I do have the following comments/questions about aspects of this study. This paper also discusses very important points relative to the significance, difficulty, and limitations of evaluation of post-mortem body condition in killer whales. The discussion about potential fisheries interaction with killer whales is also important.

Based on how the data was collected, I am doubtful that the data fit the assumptions of the ANCOVA (normality, homogeneity of variance and randomized independent samples- samples were independent, not random) used to determine differences between body length and blubber thickness measurements relative to location. Therefore, I recommend an alternative procedure (e.g. generalized linear model, data transformations, nonlinear model) and for other results a non-parametric one-sided t-test alternative (wilcoxon signed rank test).Can you also explain figure 10 in a bit more detail? I did not see the figure title or legend.

Also, it would be helpful, in addition to the large table of observations, to summarize some of the data in a graph or other type of data visualization(e.g. cause or causes of death and any significant associations with other factors).

6. PLOS authors have the option to publish the peer review history of their article (what does this mean?). If published, this will include your full peer review and any attached files.

Reviewer #1: No

Reviewer #2: No

---

## [Author Response · Author response to Decision Letter 0]

12 Oct 2020

Dear Dr. Caballero,

Thank you for reviewing our manuscript PONE-D-20-23093, “Pathology findings and correlation with body condition index in stranded killer whales (Orcinus orca) in the northeastern Pacific and Hawaii from 2004 and 2013.” We were encouraged to hear that both reviewers were very positive regarding this manuscript and its results and they both agree this is an important contribution to the field and will help many researchers working on this and similar species.

All of the comments were very helpful and improved the quality of the manuscript. As requested, we have addressed all concerns raised by both Reviewers. We have uploaded a marked-up copy of the manuscript that highlights changes made to the original version and labeled it “Revised Manuscript with Track Changes.” Also, we have uploaded an unmarked version without tracked changes as a separate file labeled “Manuscript.” We have proofread the manuscript closely for mistakes and I hope you will find this revised manuscript suitable for publication.

Here is a detailed description of each point brought up during the review and what we did to address it:

Reviewer #1:

Line 162 (BCI equation, now line 165) - 'enterior' should be 'anterior'

- corrected

Line 190-191 (now line 201)- this should either be 'association .... was evaluated' or 'associations.... were evaluated'

- corrected

Line 231(now line 267)- add 'which' ("was an offshore calf which had a hiatal hernia...")

- done

Line 258 (now line 297) - should be bone remodeling rather than boney

- corrected

Line 392 (now line 436) - add 'with' or 'in' (was found with moderate...)

- done

Line 395 (now line 439). Meconium staining of what? Also please detail the signs of fetal distress that you screened for but were not present

- we removed this sentence from the manuscript “There was no indication of meconium staining or in utero fetal distress.”

Line 677 - Please remove the comma after peritonitis

- done

Line 678 (now line 763)- Bacteria should be bacterial

- fixed

Line 694 (now line 780) - I think this should be cytoplasmic rather than cytoplastic?

- fixed

Line 716 (now line 804) - 'concerns' is not a particularly scientific word. Is there a more precise subheading that could be used here?

- changed “concerns” to say “pathology”

Line 718 and 719 (now line 807) - 'this' lesion is used in the first sentence and then 'these lesions' in the following one. Please be consistent

- corrected all to “this lesion”

Line 719 – 722 (now lines 808-814) - Please include references for pathogenesis of myocardial fibrosis. The sentence that begins with "The pathogenesis" requires some work. What do you mean by 'long past inflammation'. The word 'infectious' can be removed (pathogens is sufficient). 'Diving issues' is not a scientific term; please use more precise language here. Finally, the absence of cardiomyopathy, pulmonary congestion and hepatic congestion does not rule out the possibility of non-congestive cardiac abnormalities such as arrhythmias, as has been described in association with myocardial fibrosis in humans and domestic species.

- we included two citations for pathogenesis of myocardial fibrosis (citations 54 and 55)

- we edited “long-past” to say “resolving and possibly progressive fibrosis”

- we removed the word “infectious” before pathogens

- “diving issues such as” were replaced with “gas bubble disease” 

- as suggested, we added the final concluding comment, “however, the lack of these lesions does not preclude the possibility of an arrythmia or other non-congestive cardiac abnormalities.” 

A final general comment. I found the conclusions about blubber depth and BCI particularly interesting and relevant to researchers working on cetacean health and disease and wondered if it would be worth re-emphasizing these findings in the concluding paragraph.

- as suggested, we added several sentences to re-emphasize the findings related to blubber depth and BCI in the concluding paragraph (lines 922-931).

Reviewer #2:

Based on how the data was collected, I am doubtful that the data fit the assumptions of the ANCOVA (normality, homogeneity of variance and randomized independent samples- samples were independent, not random) used to determine differences between body length and blubber thickness measurements relative to location. Therefore, I recommend an alternative procedure (e.g. generalized linear model, data transformations, nonlinear model) and for other results a non-parametric one-sided t-test alternative (Wilcoxon signed rank test). 

Note: edits discussed below include additions to materials and methods (lines 190-199; 203-207; 223-237), results (lines 478-491; 520-536), Fig. 9 description (lines 511-513), Fig. 10 description (line 564), Discussion (line 605; line 626; line 640; line 646; and lines 922-931).

- As suggested, we used a generalized linear model to assess relationships between body length and blubber thickness measurements for all animals combined as well as for older animals and young animals separately. All data incorporated into ANCOVA passed both normality and homogeneity of variance tests, but we had previously failed to include those details in the methods. Because of the results of the GLM, however, we assessed differences in thickness across the three blubber measurement sites separately for older animals (via ANCOVA) and young animals (via ANOVA) in this revised manuscript. Before conducting these analyses, we confirmed that data passed normality and homogeneity of variance tests. This is now stated in the methods. Finally, because both tests passed, including the test for normality, the data were not transformed.

- As the reviewer pointed out, some data are nonlinear. This is the case for the relationship between body length and BCI. It was incorrect to evaluate the data via linear regression analysis. We now evaluate the relationship between body length and BCI via Spearman Rank Order Correlation.

- Regarding the t-tests, it was an oversight to use t-tests for some of the comparisons. Data for a few of the t-tests did not pass the normality test. For these specific comparisons, the Mann-Whitney Rank Sum Test is now used. The methods were modified to specify this approach, and the results were modified to include the results of the Mann-Whitney Rank Sum Test, when appropriate. The Wilcoxon signed rank test was suggested by the reviewer, but this is actually not an appropriate test. The Wilcoxon signed-rank test is used to compare two related samples, matched samples, or repeated measurements on a single sample, which doesn’t apply in this situation. We used Mann-Whitney Rank Sum Tests to determine whether there were differences in lengths, BCI and blubber thickness measurements between animals that died from trauma and animals that died from disease or malnutrition (combined). T-tests are still used for the specific cases when the data are normal and other assumptions of t-tests are met. This is now explicitly stated in the methods and results. We reiterate that one-tailed tests were chosen because, based on previous publications, we assumed that animals that died from trauma would have thicker blubber and greater BCI values than those that died from disease and malnutrition. We selected one-tailed T-tests prior to conducting the statistical analyses. However, regardless of whether one-tailed or two-tailed tests are used for the comparisons, the results are still highly significant.

Can you also explain figure 10 in a bit more detail? I did not see the figure title or legend.

- Figure 10 has a lengthily description in the figure legend, however it appears on the page prior to the figure so I imagine Reviewer #2 just missed it.

Also, it would be helpful, in addition to the large table of observations, to summarize some of the data in a graph or other type of data visualization(e.g. cause or causes of death and any significant associations with other factors).

- Good idea. We added new summary text (lines 237-242) and Table 3 (line 247) to summarize the 62 significant pathologic findings by age class and help the reader better visualize them in a small table. 

- Additionally, we added an additional reference to Table 1 in the materials and methods (line 124) and alphabetized and corrected the terms used for cause of death in Table 1 (lines 127-129).

Note: in reviewing the manuscript one final time, we noted that the pathogenesis described on lines 274-275 was not clear, so we altered the text slightly to clarify. Also, 

• we added an acknowledgment line of thanks to G. Anchor for help with the GLM (line 960)

• we corrected the spelling of cholangiohepatitis in Table 1 for ID 20040503

• we added “cm” after the measurement 501 on line 159

All co-authors have approved these revisions. Again, I would like to reiterate that the peer-review process at PLOS ONE has improved the quality of this manuscript and I appreciate the thoughtful and helpful comments provided. I hope you will find this manuscript acceptable for publication. Please do not hesitate to contact me by email or by phone if you have any questions or need additional information.

---

## [Decision Letter · Decision Letter 1]

4 Nov 2020

Pathology findings and correlation with body condition index in stranded killer whales (Orcinus orca) in the northeastern Pacific and Hawaii from 2004 and 2013

PONE-D-20-23093R1

Dear Dr. Gaydos,

We’re pleased to inform you that your manuscript has been judged scientifically suitable for publication and will be formally accepted for publication once it meets all outstanding technical requirements.

Kind regards,

Susana Caballero, PhD

Academic Editor

PLOS ONE

Additional Editor Comments (optional):

Well done, excellent paper, will be a very good addition to marine mammal literature

Reviewers' comments:

Reviewer's Responses to Questions

**Comments to the Author**

1. If the authors have adequately addressed your comments raised in a previous round of review and you feel that this manuscript is now acceptable for publication, you may indicate that here to bypass the “Comments to the Author” section, enter your conflict of interest statement in the “Confidential to Editor” section, and submit your "Accept" recommendation.

Reviewer #1: All comments have been addressed

Reviewer #2: All comments have been addressed

2. Is the manuscript technically sound, and do the data support the conclusions?

Reviewer #1: (No Response)

Reviewer #2: Yes

3. Has the statistical analysis been performed appropriately and rigorously? 

Reviewer #1: (No Response)

Reviewer #2: Yes

4. Have the authors made all data underlying the findings in their manuscript fully available?

Reviewer #1: (No Response)

Reviewer #2: Yes

5. Is the manuscript presented in an intelligible fashion and written in standard English?

Reviewer #1: (No Response)

Reviewer #2: Yes

6. Review Comments to the Author

Reviewer #1: (No Response)

Reviewer #2: All comments have been adequately addressed. This article is an important addition to the marine mammal disease literature.

7. PLOS authors have the option to publish the peer review history of their article (what does this mean?). If published, this will include your full peer review and any attached files.

Reviewer #1: No

Reviewer #2: No

---

## [Editor Report · Acceptance letter]

9 Nov 2020

PONE-D-20-23093R1 

Pathology findings and correlation with body condition index in stranded killer whales *(Orcinus orca)* in the northeastern Pacific and Hawaii from 2004 and 2013 

Dear Dr. Gaydos:

I'm pleased to inform you that your manuscript has been deemed suitable for publication in PLOS ONE. Congratulations! Your manuscript is now with our production department. 

Kind regards, 

on behalf of

Dr. Susana Caballero 

Academic Editor

PLOS ONE